



# Exploring the sensitivity of atmospheric nitrate concentrations to nitric acid uptake rate using the Met Office's Unified Model

Anthony C. Jones[1], Adrian Hill[1], Samuel Remy[2], N. Luke Abraham[3,4], Mohit Dalvi[1], Catherine Hardacre[1], Alan J. Hewitt[1], Ben Johnson[1], Jane P. Mulcahy[1], and Steven T. Turnock[1]

[1] Met Office, Fitzroy Road, Exeter, EX1 3PB, UK
[2] HYGEOS, Lille, France
[3] National Centre for Atmospheric Science, U.K.
[4] Department of Chemistry, University of Cambridge, Lensfield Road, Cambridge, CB2 1EW, U.K.

*Correspondence to*: Anthony C. Jones (anthony.jones@metoffice.gov.uk)

**Abstract.**

Ammonium nitrate is a major aerosol constituent over many land regions and contributes to air pollution episodes, ecosystem destruction, regional haze, and aerosol-induced climate forcing. Many climate models that represent ammonium nitrate assume that the ammonium-sulphate-nitrate chemistry reaches thermodynamic equilibrium instantaneously without considering kinetic limitations on condensation rates. The Met Office's Unified Model (UM) is employed to investigate the sensitivity of

ammonium nitrate concentrations to the nitric acid uptake coefficient ($\gamma$) in a newly-developed nitrate scheme in which first order condensation theory is utilised to limit the rate at which thermodynamic equilibrium is attained. Two values of $\gamma$ representing fast ($\gamma = 0.193$) and slow ($\gamma = 0.001$) uptake rates are tested in 20-year global UM integrations. The global burden of nitrate associated with ammonium in the 'fast' simulation (0.11 Tg[N]) is twice as great as in the 'slow' simulation (0.05 Tg[N]), while the top-of-the-atmosphere radiative impact of representing nitrate is -0.19 Wm$^{-2}$ in the 'fast' simulation and -

0.07 Wm$^{-2}$ in the 'slow' simulation. In general, the 'fast' simulation exhibits better spatial correlation with observed nitrate concentrations while the 'slow' simulation better resolves the magnitude of concentrations. Local near-surface nitrate concentrations are found to be highly correlated with seasonal ammonia emissions suggesting that ammonia is the predominant limiting factor controlling nitrate prevalence. This study highlights the high sensitivity of ammonium nitrate concentrations to nitric acid uptake rates and provides a mechanism for reducing nitrate concentration biases in climate model simulations. The

new UM nitrate scheme represents a step-change in aerosol modelling capability in the UK across weather and climate timescales.

## 1 Introduction

Air pollution poses a significant hazard to human-health and to the environment world-wide. In 2016, 90 % of the global population were exposed to pollutant concentrations in excess of World Health Organisation (WHO) defined safe levels,

resulting in ~7 million premature deaths (WHO, 2020). Specific human-health conditions arising from air pollution exposure



include lung cancer and cardiopulmonary disease, and deleterious impacts also extend to ecosystems (e.g. eutrophication, loss of biodiversity, acid deposition), building and infrastructure erosion, and impaired atmospheric visibility and regional haze (Kucera and Fitz, 1995; Monks *et al.*, 2009; Lovett *et al.,* 2009; Xu *et al.*, 2019). Solid or liquid particulate matter (PM) is a significant component of air pollution and particles with diameters less than 2.5 µm (i.e. $PM_{2.5}$) are particularly harmful to

human health. Lelieveld *et al.* (2015) estimate $PM_{2.5}$ related global mortality to be 3.3 million deaths $yr^{-1}$ in 2010, far greater than the second deadliest air pollutant, ozone ($O_3$, 142 thousand deaths $yr^{-1}$). Sources of air pollution differ with region; in North Africa and the Middle East, the predominant source is naturally emitted dust; in Europe, agricultural activity; and in South East Asia, residential energy production (Lelieveld *et al.*, 2015).

Secondary inorganic ammonium ($NH_4$), sulphate ($SO_4$), and nitrate ($NO_3$) aerosol form a major part of $PM_{2.5}$ composition in the Northern Hemisphere (Jiminez *et al.*, 2009). Ammonium is predominantly emitted as ammonia ($NH_3$) gas by agricultural sources such as mineral fertilizer application and volatilization of livestock manure; biomass burning and from ocean emission (Bauer *et al.*, 2016). $NH_3$ emissions from agriculture have dramatically increased since the discovery of the Haber-Bosch process for extracting reactive nitrogen from its stable atmospheric form ($N_2$) in the early 20[th] Century. The

corresponding $NH_3$-based fertilizer revolution led to significantly enhanced global food production and a population explosion from 2 billion to 7 billion people (Smith *et al.*, 2020). However, reactive nitrogen deposition from fertiliser usage is now 20-fold higher than it was before the industrial revolution (Xu *et al.*, 2019). $NO_3$ is formed from atmospheric nitric acid ($HNO_3$), itself an oxidation product of nitrogen oxides ($NO_x = NO + NO_2$). $NO_x$ is primarily emitted from anthropogenic fossil fuel burning (21-28 Tg N $yr^{-1}$) but has natural sources including soil emissions, biomass burning and

lightning (12-35 Tg N $yr^{-1}$) (Seinfeld and Pandis, 1998; Vinken *et al.*, 2014). $NO_x$ exacerbates air pollution via two pathways – by $NO_3$ aerosol production and by net $O_3$ production in the presence of sunlight and volatile organic compounds (VOCs) (Crutzen, 1970). $SO_4$ is the oxidation product of sulphur dioxide ($SO_2$), which is primarily emitted by anthropogenic processes such as fossil fuel combustion, petroleum refining, and metal smelting (Zhong *et al.*, 2020). Natural $SO_2$ sources include volcanic degassing and the oxidation of reduced natural sulphurous compounds such as dimethyl sulphide (DMS)

(Carn *et al.*, 2017). Global anthropogenic $SO_2$ emissions have steadily declined from a peak of ~70 Tg S $yr^{-1}$ in the 1980s to ~52 Tg S $yr^{-1}$ in 2014 owing to clean air regulation instigated to mitigate adverse $SO_2$ impacts such as acid rain (Mcduffie *et al.*, 2020; Zhong *et al.*, 2020). Global $NO_x$ emissions peaked in 2010 at ~40 Tg N $yr^{-1}$, with emissions growth from 1990-2010 driven by rapid industrialisation in Asia and intensified international shipping, but have decreased by 7 % over the 2010s owing primarily to traffic emission control measures in China (Mcduffie *et al.*, 2020).


In the troposphere, $NO_x$ is involved in a complex diurnal photochemical cycle involving VOCs and $O_3$. The dominant $NO_x$ removal mechanism during daytime is via oxidation by hydroxyl (OH) radicals to form $HNO_3$ (Seinfeld and Pandis, 1998). At night-time, $NO_2$ is unable to photolyze and the dominant $NO_x$ removal mechanism is via reaction with $O_3$ to produce the $NO_3$ radical; which further reacts with $NO_2$ to form dinitrogen pentoxide ($N_2O_5$); which heterogeneously reacts with water ($H_2O$)



to produce $HNO_3$ (Atkinson, 2000). $HNO_3$ is abundantly soluble and rapidly dissolves in water droplets or is neutralised by $NH_3$ to form aerosol. $SO_2$ is oxidised in the aqueous phase by dissolved oxidants such as $O_3$ and hydrogen peroxide ($H_2O_2$), and in the gas phase by OH, to form non-volatile $H_2SO_4$ (Seinfeld and Pandis, 1998). $H_2SO_4$ and $HNO_3$ react with $NH_3$ to produce ammonium sulphate (($NH_4)_2 \cdot SO_4$) and semi-volatile ammonium nitrate ($NH_4 \cdot NO_3$) aerosol respectively, with $H_2SO_4$ neutralisation taking precedence owing to the lower vapor pressure of $H_2SO_4$ over $HNO_3$ (Hauglustaine *et al.*, 2014). In a

slower process, $HNO_3$ also condenses irreversibly onto existing dust and sea-salt aerosols, forming calcium nitrate ($Ca \cdot (NO_3)_2$) and sodium nitrate ($Na \cdot NO_3$) salts respectively (Li and Shao, 2009). Owing to the prevalence of anthropogenic $NO_x$ and $NH_3$ sources, particulate $NO_3$ is a major component of urban air pollution. For example, in Europe $NO_3$ constitutes 17 % of urban $PM_{2.5}$ aerosol by mass (Putaud *et al.*, 2004), while $NH_4 \cdot NO_3$ can comprise 75 % of $PM_{2.5}$ in air pollution events in Salt Lake City (Womack *et al.*, 2019).


Nitrate aerosol has an enigmatic history within the climate modelling community owing to the complexity of modelling $HNO_3$ neutralisation by $NH_3$ and the semi-volatility of $NH_4 \cdot NO_3$ aerosol. In the inorganic aerosol system, gaseous and particulate equilibria are reached at different rates due to evolving temperature and acidity constraints, and the variability in gaseous uptake with particle size (Myhre *et al.*, 2006; Benduhn *et al.*, 2016). Although complex dynamical and 'hybrid dynamical'

schemes that fully or partially resolve the chemistry of inorganic aerosol exist (e.g. Jacobson, 1997; Feng and Penner, 2007; Zaveri *et al.*, 2007; Benduhn *et al.*, 2016; Xu and Penner, 2012), they remain computationally expensive – owing to the numerical stiffness of the inorganic system – when compared to schemes that assume thermodynamic equilibrium is reached instantaneously (Nenes *et al.*, 1998). Most of the current crop of nitrate-resolving global climate models (GCMs) and some regional climate models (RCMs) assume the instantaneous thermodynamic equilibrium approximation (Liao *et al.*, 2003;

Myhre *et al.*, 2006; Bauer *et al.*, 2007; Bellouin *et al.*, 2011; Hauglustaine *et al.*, 2014; Paulot *et al.*, 2016; Bian *et al.*, 2017; Remy *et al.*, 2019). Dynamical models have the advantage of capturing natural phenomena where the inorganic aqueous system is outside of (or slow to reach) equilibrium, for instance in low gas concentrations, low temperatures, high Relative Humidities (RH), and for condensation onto coarse particles (Wexler and Seinfeld, 1990; Benduhn *et al.*, 2016). Thermodynamic equilibrium models typically overestimate the fraction of $NO_3$ in the coarse mode, for example, in one study $NO_3$ associated

with fine mode $NH_4 \cdot NO_3$ was underestimated by 25 % compared to a hybrid-dynamical model (Feng and Penner, 2007). However, the additional computational expense of using dynamical approaches has motivated the climate modelling community to seek pragmatic solutions to represent $NH_4$ and $NO_3$ aerosol in GCMs and RCMs.

GCM simulations suggest that the present-day direct radiative forcing from $NO_3$ (global mean $\approx$ -0.1 $Wm^{-2}$) amounts to a

quarter of the $SO_4$ forcing on a global-mean basis (Myhre *et al.*, 2006; Bauer *et al.*, 2007; Bellouin *et al.*, 2011; Hauglustaine *et al.*, 2014). $NO_3$ aerosol burdens are widely projected to increase over the 21$^{st}$ century as a result of stricter $SO_2$ emissions regulations, which would reduce the $SO_4$ available for neutralisation and concomitantly liberate $NH_4$ for $NH_4 \cdot NO_3$ formation (Bauer *et al.*, 2007; Bellouin *et al.*, 2011). Consequently, $NO_3$ may become the dominant aerosol species in terms of radiative





and urban air pollution impact by the end of the century, depending on future emissions of $SO_2$, $NH_3$ and $NO_x$ (Hauglustaine

*et al.*, 2014). Such $NO_3$ concentration enhancements may be effectively mitigated on a regional basis by judicious regulation targeting anthropogenic $NH_3$ and/or $NO_x$ emissions (Bauer *et al.*, 2016). However, climate models disagree as to whether near-surface $NO_3$ concentrations will increase or decrease in future climate, and on the correct partitioning between $NO_3$ in the fine mode (associated with $NH_4$) and coarse mode (associated with dust and sea-salt) (Bian *et al.*, 2017). Many of the uncertainties in $NH_4$ and $NO_3$ projections emanate from different treatments of the $HNO_3$ and $NH_3$ gases in models, with Bian *et al.* (2017)

highlighting wet deposition as a particularly sensitive process. Additionally, the vertical distributions of $HNO_3$ and $NH_3$ are poorly constrained by observations which adds to uncertainty in $NH_4$ and $NO_3$ projections (Paulot *et al.*, 2016).

The emerging conclusion from observations and from the burgeoning literature on nitrate modelling is that ammonium nitrate poses an increasingly significant health hazard through urban air pollution (e.g. DEFRA, 2012) and via deposition to nitrogen-

saturated ecosystems (Li *et al.*, 2016); and potentially could become a major climate forcing agent as $SO_4$ concentrations wane (Hauglustaine *et al.*, 2014). The impetus for explicitly representing $NH_4$ and $NO_3$ in GCMs is clear, even by using simple thermodynamic equilibrium approaches which bypass temporal nuances in the gas-particulate partitioning. The Met Office Unified Model (UM) has previously incorporated a thermodynamic-equilibrium ammonium nitrate scheme in the CMIP5-generation climate model Hadley Centre Global Environment Model version 2 (HadGEM2-ES) (Bellouin *et al.*, 2011). This

nitrate scheme – developed within the single-moment Coupled Large-scale Aerosol Simulator for Studies in Climate (CLASSIC) aerosol framework (Bellouin *et al.*, 2007) – continues to be utilised for online air quality forecasts across the UK in the operational Air Quality Unified Model (AQUM) (Savage *et al.*, 2013). However, the CMIP6-generation state-of-the-art United Kingdom Earth System Model version 1 (UKESM1) which incorporates the Global Atmosphere model vn7.1 (GA7.1) (Walters *et al.*, 2019) replaced the single-moment CLASSIC aerosol scheme with the double-moment Global Model of Aerosol

Processes modal (GLOMAP-mode) scheme which currently omits ammonium nitrate (Mann *et al.*, 2010; Mulcahy *et al.*, 2020). Mulcahy *et al.* (2020) attributed a negative bias in aerosol optical depth and mass burden over northern hemisphere continents in UKESM1 to the missing $NH_4$ and $NO_3$. The hybrid-dynamical nitrate scheme developed by Benduhn *et al.* (2016) in the standalone GLOMAP-mode model is yet to successfully transition to the UM. This has provided the Met Office with fresh impetus to develop a simplified thermodynamic equilibrium nitrate scheme within the GLOMAP-mode framework for

interim use in the UM and possible implementation in future generations of UKESM, in order to fill the $NH_4$ and $NO_3$ shaped void. The nitrate scheme may garner further utility if AQUM or its successor transitions to utilising the GLOMAP-mode aerosol scheme rather than CLASSIC (Hemmings and Savage, 2018).

In this paper we describe the development and testing of a simple thermodynamic equilibrium nitrate scheme in the UM and

explore the sensitivity of the scheme to a key parameter that is poorly constrained by observations – the $HNO_3$ uptake coefficient on aerosol surfaces (γ). Specifically, most models assume that $NH_4 \cdot NO_3$ concentrations reach thermodynamic equilibrium instantaneously without considering kinetic limitations on the condensation of $HNO_3$ or $NH_3$ onto existing aerosol





particles, as is considered here. The UM nitrate scheme reported here comprises fine $NH_4$ and $NO_3$ aerosol in the Aitken, accumulation, and coarse soluble modes and coarse $NO_3$ representing $NO_3$ aerosol associated with dust and sea-salt in the

accumulation and coarse soluble modes. The scheme was originally developed by Hauglustaine *et al*. (2014) for use in the LMDZ-INCA climate model, then adapted for ECMWF's version of GLOMAP-mode by Remy *et al.* (2019), following which it was kindly provided to the Met Office for adaptation to the UM. In Section 2.1, we describe the configuration of the UM used to test the new nitrate scheme. In Section 2.2, we describe the thermodynamic equilibrium nitrate model in detail. In Section 2.3 we describe the simulations performed in this study. In Section 3, we evaluate the model using surface and satellite

observations and investigate the sensitivity of the model to perturbations to a key parameter – the $HNO_3$ uptake coefficient ($\gamma$) – in a manner analogous to Bauer *et al*. (2004). In Section 4, we discuss the utility of the nitrate scheme and provide a roadmap for future development and integration within UKESM.

## 2 Methods

### 2.1 The Met Office Unified Model (UM)

The nitrate scheme was originally developed using the UM with the science configurations Global Atmosphere vn7.1 (GA7.1) and Global Land vn7.0 (GL7.0) (Walters *et al,* 2019). Although the UM can be run at various resolutions, the resolution used here is the climate configuration N96L85, i.e. 1.875° longitude by 1.25° latitude with 85 vertical levels up to a model lid at 80 km, with 50 levels below 18 km altitude, and a model timestep of 20 mins (Walters *et al*, 2019). The UM's dynamical core, Even Newer Dynamics for General atmospheric modelling of the environment (ENDGame), is a semi-implicit and semi-

Lagrangian solver for the non-hydrostatic deep-atmosphere equations of motion (Wood *et al.*, 2014). Mass conservation of moist tracers is achieved using the Optimised Conservative Filter scheme (Zerroukat and Allen, 2015). Atmospheric radiative transfer is modelled assuming the two-stream approximation, with 6 wavebands in the shortwave spectrum and 9 wavebands in the longwave spectrum (Manners *et al*., 2015). Large-scale cloud is modelled with the Prognostic Cloud fraction and Prognostic Condensate (PC2) scheme (Wilson *et al.*, 2008) and cloud microphysics uses the single-moment scheme based on

Wilson and Ballard (1999) with extensive modifications. Full details of GA7.1/GL7.0 are provided in Walters *et al*. (2019) and Mulcahy *et al*. (2018).

In the model configuration used here, GA7.1 includes the United Kingdom Aerosol and Chemistry (UKCA) model which simulates atmospheric composition in the Earth System, with UKCA chemistry called once per model hour in N96L85, 

although emissions are evaluated every model timestep (Archibald *et al*., 2020). UKCA is coupled to the GLOMAP-mode aerosol scheme, permitting a holistic and prognostic treatment of aerosol and chemical processes over the entire atmosphere (Mann *et al*., 2010; Mulcahy *et al*., 2020). The coupled UKCA and GLOMAP-mode model is widely referred to as UKCA-mode. The Met Office's Hadley Centre Global Climate version 3.1 (HadGEM3-GC3.1) model – the physical basis of UKESM1 – uses a simplified UKCA chemistry configuration with important offline oxidants ($O_3$, OH, $NO_3$, $HO_2$) provided as monthly





mean climatologies (Walters *et al.*, 2019; Mulcahy *et al.*, 2020). This is of insufficient complexity for ammonium nitrate aerosol, given the importance of missing gases (i.e. $HNO_3$, $NH_3$ and precursors) and chemical reactions. Instead the combined Stratospheric-Tropospheric version 1.0 (StratTrop 1.0) chemistry scheme, which is included in UKESM1 (Sellar *et al.*, 2019) and described in detail by Archibald *et al.* (2020), is utilised here. Although not mentioned in Archibald *et al.* (2020), gaseous ammonia ($NH_3$) is a passive tracer in StratTrop1.0, while gaseous nitric acid ($HNO_3$) is the product of various atmospheric

chemical reactions (see Table S1 in the Supplement). StratTrop1.0 simulates the chemical cycles of $O_x$, $HO_x$, $NO_x$, and halogenic compounds; the oxidation of various organic compounds; and heterogenous chemistry on Polar Stratospheric Cloud (PSCs), Nitric Acid Trihydrate (NAT) and tropospheric aerosols, with 84 species and 291 chemical reactions. Further details of the nitrogen chemistry in StratTrop1.0 is provided in Section 2.2. Interactive photolysis in UKCA is modelled with the Fast-JX scheme, which covers the wavelength range 177-750 nm (Telford *et al.*, 2013). Gaseous wet deposition follows the effective

Henry's law approximation described in Giannakopous *et al.* (1999) while dry deposition is treated with a resistance type model (O'Connor *et al.*, 2014). Gaseous dissolution in cloud droplets is modelled using the effective Henry's law framework, with UKCA uniformly assuming a fixed cloud water pH of 5.0. Values required to calculate the effective Henry's law coefficients are specified as $K_H(298K) = 2.1 \times 10^5$, 1.23, and $1 \times 10^6$ for $HNO_3$, $SO_2$ and $NH_3$ respectively, and $-\Delta H/R = 8700$, 3020, and 0 $K^{-1}$ for $HNO_3$, $SO_2$ and $NH_3$ respectively (Archibald *et al.*, 2020). The values for $NH_3$ are comparable to AeroCom

phase III values given in Table 5 in Bian *et al.* (2017).

    GLOMAP-mode is a prognostic double-moment aerosol scheme that carries aerosol mass and number concentrations in 4 soluble lognormal modes spanning sub-micron to super-micron sizes (nucleation, Aitken, accumulation, and coarse), as well an insoluble Aitken mode (see Table 1) (Mann *et al.*, 2010; Mulcahy *et al.*, 2020). The variable size distribution allows the

median dry radius to evolve within prescribed size brackets, while the lognormal standard deviation or 'mode width' is held fixed. GA7.1's default GLOMAP-mode configuration includes the aerosols sulphate ($SO_4$), black carbon (BC), organic matter (OM), and sea-salt (SS), with species in each mode treated as an internal mixture. Mineral dust is represented in GA7.1 by the CLASSIC 6 bin scheme described by Woodward *et al.* (2001). Aerosol processes in GLOMAP-mode include binary homogeneous $SO_4$ nucleation, mode-merging, cloud processing, condensational uptake of $H_2SO_4$ gas and secondary organic

vapour from UKCA, and coagulation within and between modes. Aerosol water content is simulated prognostically using the Zdanovskii-Stokes-Robinson (ZSR) method, allowing for a more accurate representation of aerosol-cloud interactions and aerosol radiative impact than in CLASSIC. The direct aerosol radiative effect is modelled using UKCA-Radaer, which utilises pre-determined look-up tables of Mie extinction parameters based on aerosol size and composition (Bellouin *et al.*, 2013). Cloud condensation nuclei are converted to cloud droplets using UKCA-Activate which is based on the Abdul-Razzak and

Ghan (2000) parameterization, while autoconversion of cloud droplets to rainwater follows Khairoutdinov and Kogan (2000) (Boutle *at al.*, 2014; West *et al.*, 2014; Mulcahy *et al.*, 2018). Aerosol removal processes treated by UKCA-mode include dry deposition and sedimentation, in-cloud scavenging by the autoconversion of cloud to rainwater; beneath cloud scavenging by rain and snow impaction; and convective plume scavenging (Mann *et al.*, 2010, Mulcahy *et al.*, 2020).



| Aerosol Mode | Geometric mean diameter $\bar{D}$ (nm) | $\sigma_g$ | Species |
|---|---|---|---|
| Nucleation soluble | 1 – 10 | 1.59 | SO₄, OM |
| Aitken soluble | 10 – 100 | 1.59 | SO₄, BC, OM, NH₄, NO₃ |
| Accumulation soluble | 100 – 1000 | 1.4 | SO₄, BC, OM, SS, NH₄, NO₃, hetNO₃ |
| Coarse soluble | > 1000 | 2.0 | SO₄, BC, OM, SS, NH₄, NO₃, hetNO₃ |
| Aitken insoluble | 10 - 100 | 1.59 | BC, OM |


**Table 1: Properties of the aerosol size distribution in the nitrate UKCA-mode setup including the size range of the modes, the geometric standard deviation, and the permitted aerosol species in each mode. Species include sulphate (SO₄), black carbon (BC), organic matter (OM), sea-salt (SS) and the newly added ammonium (NH₄), nitrate (NO₃) and coarse nitrate (hetNO₃). Adapted from Table 1 in Mulcahy *et al*. (2020).**


## 2.2 Nitrate model

In addition to the standard aerosols in GA7.1 - SO₄, BC, OM, and SS - ammonium (NH₄), nitrate (NO₃) and coarse nitrate (herein denoted hetNO₃ for convenience) are added to a new UKCA-mode setup which comprises 28 aerosol tracers in total (Table 1). NH₄ and NO₃ mass is emitted into the Aitken and accumulation soluble modes and may be transferred to the coarse

soluble mode via aerosol processing, while hetNO₃ is limited to the accumulation and coarse soluble modes. Nitrate chemistry is evaluated once per model timestep within the UKCA emissions-control routine. The nitrate model closely follows Hauglustaine *et al*. (2014) and Remy *et al*. (2019) with subtle yet important differences. An exhaustive step-by-step methodology is provided in the Supplement (Sections S1 and S2) and outlined below.

### 2.2.1 Ammonium nitrate production

Fine mode ammonium nitrate production is evaluated before the condensation of HNO₃ onto coarse aerosols (e.g. sea-salt and dust) because smaller particles generally reach thermodynamic equilibrium faster (Hauglustaine *et al*., 2014; Benduhn *et al*., 2016). Firstly, the sulphate neutralisation state ($\Gamma_{SO_4}$) is determined from the total moles of ammonia ($T_A = \{NH_3\} + \{NH_4^+\}$) and total moles of sulphate ($T_S = \{SO_4\}$) using Eq. 1 (Metzger *et al*., 2002).

$$\Gamma_{SO_4} = \begin{cases} 2 & 2\,T_S < T_A & 2NH_3 + H_2SO_4 \rightarrow (NH_4)_2SO_4 \\ 1.5 & T_S < T_A < 2\,T_S & 3NH_3 + 2H_2SO_4 \rightarrow (NH_4)_3H(SO_4)_2 \\ 1 & T_A < T_S & NH_3 + H_2SO_4 \rightarrow (NH_4)HSO_4 \end{cases} \quad [1]$$



The moles of ammonia available for neutralisation of $HNO_3$ following the irreversible production of ammonium sulphate is then $T_A^* = T_A - \Gamma_{SO_4} T_S$. If all free ammonia is consumed by the neutralisation of $SO_4$ ($T_A^* = 0$) then no new nitrate is formed. However, if ammonia is available ($T_A^* > 0$) then the equilibrium constant ($K_p$) of the ammonia-nitrate system (Eq. 2) is

determined using the parameterisation of Mozurkewich (1993) (see Section S1 in the Supplement). In this formulation, $K_p$ is solely a function of temperature and Deliquescence Relative Humidity (DRH), with DRH following the parameterisation of Seinfeld and Pandis (1998).

$$HNO_3 + NH_3 \overset{K_p}{\leftrightarrow} NH_4NO_3 \qquad [2]$$


The equilibrium concentration of ammonium nitrate is then calculated using the formulation from Seinfeld and Pandis (1998). Letting $T_N$ denote the total molar concentration of nitrate ($T_N = \{HNO_3\} + \{NO_3^-\}$), if $T_A^* T_N > K_p$ then the molar concentration of ammonium nitrate aerosol at equilibrium is calculated using Eq. 3. Else if free ammonia or nitrate concentrations are limited such that $T_A^* T_N \leq K_p$ or $T_A^* = 0$ then all existing ammonium nitrate aerosol evaporates and the corresponding mass is

transferred to the gas phase $HNO_3$ and $NH_3$.

$$\{NH_4NO_3\}_{eq} = \frac{1}{2}\left[T_A^* + T_N - \sqrt{(T_A^* + T_N)^2 - 4(T_N T_A^* - K_p)}\right] \qquad [3]$$

The ammonia-nitrate system may not reach equilibrium within a standard GCM timestep owing to transport limitations

between the gas and aerosol phases (Wexler and Seinfeld, 1990). The time taken to reach equilibrium depends on ambient temperature and RH, and the aerosol size and uptake coefficient ($\gamma$), where the uptake coefficient is defined as the number of gas molecules condensing on a particle divided by the number impacting onto the particle surface. Ackerman *et al.* (1995) find that equilibration time ($\tau$) may range from ~2 minutes for particles with diameters of 0.1 µm to ~1 hour for diameters of 0.5 µm, depending on the uptake rate. Remy *et al.* (2019) assumed a globally uniform equilibration time of $\tau = 2$ minutes in their

nitrate model. Here the uptake rate $k_{HNO3}$ is determined for each aerosol mode online (Aitken, accumulation, and coarse soluble) using the first order uptake theory of Schwartz (1986) and by applying the Fuchs and Sutugin (1970) correction factors for molecular effects and for limitations in interfacial mass transport (Eqs 4-7).

$$D_g = \frac{3}{8A_c\rho_a d_a^2}\left[\frac{m_a R_a T}{2\pi} \times \frac{m_a + m_{HNO3}}{m_{HNO3}}\right]^{\frac{1}{2}} \qquad [4]$$

$$\lambda = \frac{3D_g}{\upsilon} = \frac{3D_g}{\sqrt{\frac{8R_a T}{\pi m_{HNO3}}}} \qquad [5]$$




$$K_n = \frac{2\lambda}{D} \qquad\qquad [6]$$

$$k_{HNO3} = \frac{2\pi D D_g}{1 + \frac{4K_n}{3\gamma} \times \left(1 - \frac{0.47\gamma}{1+K_n}\right)} \qquad [7]$$

Equations 4-7 determine the molecular diffusivity coefficient ($D_g$, m$^2$s$^{-1}$), the mean free path ($\lambda$, m), the Knudsen number ($K_n$),

and the modal condensation or uptake rate ($k_{HNO3}$, m$^3$s$^{-1}$) respectively. Constants in the algorithm include the Avogadro

constant $A_c = 6.022 \times 10^{23}$ mol$^{-1}$, the gas constant of dry air $R_a = 8.314$ J mol$^{-1}$ K$^{-1}$, the molar mass of dry air $m_a = 0.029$

kg mol$^{-1}$, the molar mass of HNO$_3$ $m_{HNO3} = 0.063$ kg mol$^{-1}$, the molecular diameter of dry air molecules $d_a = 4.5 \times 10^{-10}$ m,

and the reactive uptake coefficient ($\gamma$) for HNO$_3$. Variables in Eqs 4-7 include the air temperature $T$ (K) and air density $\rho_a$ (kg

m$^{-3}$). In Eq. 5, $v$ is the mean molecular speed (m s$^{-1}$). The total equilibration time $\tau$ (s) may then be related to the inverse of

product of the uptake rate for one particle $k_{HNO3}$ and the aerosol number concentration $N$ using Eq. 8. Note that ammonium

nitrate production is limited to the Aitken and accumulation modes in this study, which is reflected in the formulation of $\tau$.

$$\tau = \frac{1}{N_{ait}k_{HNO3,ait} + N_{acc}k_{HNO3,acc}} \qquad [8]$$

Rather than assume instantaneous thermodynamic equilibrium in the ammonia-nitrate system, the model assumes an

exponential decay of the gas phase towards equilibrium using the equilibration time $\tau$ (see Section S1.4 in the Supplement).

This approach has also been used by Ackerman *et al.* (1995), Makar *et al.* (1998), and Remy *et al.* (2019). Figure S3 in the

Supplement shows the results of applying the above algorithm for $\tau$ (with $\gamma$ set to 0.193 following Feng and Penner (2007)) to

monthly mean aerosol and meteorology output from example UM integrations. Over many land regions, $\tau$ is approximately 2

mins near the surface and increases to ~15 mins at a model level height of 3000 m. Therefore, assuming a constant value of

$\tau = 2$ mins, as assumed by Remy *et al.* (2019), may significantly overestimate the rate which the ammonia-nitrate system

approaches equilibrium, particularly at higher altitudes and over maritime regions. For example, in a 20-minute timestep the

ammonia-nitrate system would move 99.995 % of the way from initial conditions towards equilibrium with $\tau = 2$ mins

assuming exponential decay, but only 86 % of the way with $\tau = 10$ mins and 33 % of the way with $\tau = 50$ mins.


For standard atmospheric conditions ($D_g = 10^{-5}$ m$^2$s$^{-1}$ and $v = 300$ ms$^{-1}$), $k_{HNO3}$ scales approximately linearly with the

reactive uptake coefficient $\gamma$, for $\gamma$ from 0.001 to 0.2 and for particle diameters between $D = 0.1$ µm and $D = 5$ µm (Figure

S2 in the Supplement). The uptake rate increases on a particle-by-particle basis with diameter, for example, ranging from 0.2

s$^{-1}$ for $D = 0.1$ µm to 5 s$^{-1}$ for $D = 0.5$ µm when $\gamma = 0.1$ and when $k_{HNO3}$ is normalised by $N = 10^{12}$ m$^{-3}$. However,

atmospheric Aitken mode number concentrations generally exceed accumulation mode concentrations, particularly over





populous land regions and increasingly with altitude. In example UM integrations, the ratio of accumulation to Aitken uptake ($N_{acc}k_{HNO3,acc}/N_{ait}k_{HNO3,ait}$) decreases on a global-mean basis from 8 at the surface to 1.4 at a model level height of 3000 m, but is effectively unity at the surface over key NH$_3$ and NO$_x$ emitting regions such as the US, Europe and South Asia (Figure S4 in the Supplement).


Uptake rates ($k_{HNO3}$, Eq. 7) are determined for the Aitken and accumulation modes by using the modal geometric-mean dry diameters for $D$ in Eqs 4-7, which are first corrected for hygroscopic growth using the RH-dependent parameterisation of Gerber *et al.* (1985). This simplified 'modal' approach differs from Hauglustaine *et al.* (2014) who divide aerosol size modes into sub-bins. If $T_A^* T_N > K_p$ then mass is transferred from the gaseous reactants NH$_3$ and HNO$_3$ to NH$_4$ and NO$_3$ in the Aitken

and accumulation soluble modes using the above algorithm. Else if $T_A^* T_N \leq K_p$ then NH$_4$ and NO$_3$ dissociate and all NH$_4$ and NO$_3$ mass in the Aitken to coarse soluble modes is instantaneously transferred to the gas phase. Ammonium nitrate chemistry solely involves condensation and evaporation (Makar *et al.*, 1998; Benduhn *et al.*, 2016); thus, aerosol number concentrations are not altered explicitly by nitrate chemistry in our model but may change indirectly due to altered coagulation and mode-merging rates arising from the additional aerosol mass. This approach differs from Hauglustaine *et al.* (2014) and Remy *et al.*

(2019) who assume that new particles are nucleated by the production of ammonium nitrate.

### 2.2.2 Coarse nitrate production

Following NH$_4$·NO$_3$ production and the associated update to HNO$_3$ concentrations, the first order uptake parameterisation described by Eqs 4-7 is further employed to model the irreversible uptake of HNO$_3$ on sea-salt and dust to produce NaNO$_3$ (Eq. 9) and Ca(NO$_3$)$_2$ (Eq. 10) respectively (Liao *et al.*, 2003; Hauglustaine *et al.*, 2014).


$$HNO_3 + NaCl \rightarrow NaNO_3 + HCl \qquad [9]$$
$$2HNO_3 + CaCO_3 \rightarrow Ca(NO_3)_2 + H_2CO_3 \qquad [10]$$

The methodology is mostly unchanged from Hauglustaine *et al.* (2014) and Remy *et al.* (2019), with only subtle adaptations

needed to integrate the scheme within UKCA-mode. As in Hauglustaine *et al.* (2014), the HNO$_3$ uptake coefficients ($\gamma$) for dust and sea-salt are RH-dependent variables based on measurements from Fairlie *et al.* (2010) for dust and Sander *et al.* (2011) for sea-salt. Additionally, dust is assumed to uniformly constitute 5 % Ca$^{2+}$ by mass, which differs from the approach in Remy *et al.* (2019) who used a spatially hetereogeneous Ca$^{2+}$ fraction. Dust alkalinity is titrated by uptake of HNO$_3$ until the dust pH is neutralised whereupon HNO$_3$ stops condensing, while no such limitation is applied for sea-salt.


As for the first order uptake parameterisation for ammonium nitrate (Section 2.2.1) and in Remy *et al.* (2019), rather than explicitly integrating the uptake rate over the aerosol size distribution, $k_{HNO3}$ is calculated individually for sea-salt in the





accumulation and coarse modes using the modal geometric-mean diameters for *D* in Eqs 4-7, and individually for each
CLASSIC dust bin using fixed geometric mean diameters (see Section S2 in the Supplement). Sea-salt number concentrations
for the two modes are inversely determined from the sea-salt mass concentrations and the modal geometric-mean dry
diameters, which implicitly assumes that sea-salt is externally mixed with other UKCA-mode aerosols. Dust particle number
concentrations are determined from prognostic dust mass concentrations and fixed size distributions for each bin. Mapping
between CLASSIC's 6 dust bins and UKCA-mode's 2 size modes follows the approach currently used to map dust emissions
between CLASSIC and UKCA-mode, with Bin 2 and half of Bin 3 mapped to the accumulation mode and the other half of
Bin 3 and Bins 4, 5 and 6 mapped to the coarse mode. The dust and sea-salt uptake rates (*k*) multiplied by the equivalent
particle number concentrations (*N*) are then used to determine tendencies to mass concentrations of coarse $NO_3$ aerosol
(het$NO_3$), sea-salt (SS), and $HNO_3$ gas (Eqs 11-13). The constants in Eqs 11-13 include the molar mass of $Ca(NO_3)_2$
$m_{Ca(NO3)2} = 0.164$ kg mol$^{-1}$, the molar mass of $NaNO_3$ $m_{NaNO3} = 0.084$ kg mol$^{-1}$, the molar mass of $HNO_3$ $m_{HNO3} = 0.063$
kg mol$^{-1}$, and the molar mass of NaCl $m_{NaCl} = 0.05844$ kg mol$^{-1}$.


$$\frac{\Delta[\text{het}NO_3]}{\Delta t} = \left( \left( Nk_{du,ACC} + Nk_{du,COA} \right) \times \frac{m_{Ca(NO3)2}}{m_{HNO3}} + \left( Nk_{SS,ACC} + Nk_{SS,COA} \right) \times \frac{m_{NaNO3}}{m_{HNO3}} \right) \times [HNO_3] \qquad [11]$$

$$\frac{\Delta[HNO_3]}{\Delta t} = - \left( 2 \times \left( Nk_{du,ACC} + Nk_{du,COA} \right) + \left( Nk_{SS,ACC} + Nk_{SS,COA} \right) \right) \times [HNO_3] \qquad [12]$$

$$\frac{\Delta[SS]}{\Delta t} = - \left( Nk_{SS,ACC} + Nk_{SS,COA} \right) \times \frac{m_{NaCl}}{m_{HNO3}} \times [HNO_3] \qquad [13]$$

**2.2.3 Technical UM modifications**

UKCA-Radaer calculates 3D aerosol extinction properties for each lognormal mode online as a function of aerosol composition
and size, which are then utilised directly within the UM's radiative transfer code (Bellouin *et al.*, 2013). Each aerosol species
requires prescribed spectral refractive indices (RI) spanning the electromagnetic spectrum from ultra-violet (0.2 µm) to
radiowave (1 cm) wavelengths. Ammonium nitrate RIs have previously been compiled for an older generation of the UM
(HadGEM2-ES) (Bellouin *et al.*, 2011). For $NH_4 \cdot NO_3$, real and imaginary RIs for the wavelength spectrum 2-20 µm are taken
from Jarzembski *et al.* (2003), while RIs for > 20 µm are assumed to be that at 20 µm. Real RIs for 0.59-1.61 µm are from
CRC Handbook of Chemistry and Physics (Weast, 1971) and are then extended to cover the 0.1-2 µm spectrum. Imaginary
RIs for the ultraviolet and visible spectra (< 0.7 µm) are arbitrarily set to a small number assuming little absorption (1×10$^{-8}$).

To optically represent the coarse $NO_3$ aggregate het$NO_3$, $NaNO_3$ spectral RIs have been compiled from the literature for this
study. RI values are mostly from the tabulated data of Palik and Khanna (1998) and references therein for solid birefringent
$NaNO_3$ crystals. From 0.23-0.42 µm, imaginary RIs are determined by applying the Beer-Lambert law to Cleaver *et al.* (1963)





absorption coefficients and assuming a lattice thickness of 3 µm, as in Jacobson (1999). This approach is necessary to account

for the second UV absorption peak missing in the data of Palik and Khanna (1998). Imaginary RIs for wavelength spectrum

0.42-5.88 µm appear not to have been measured and are pragmatically set to the observed values for $NH_4 \cdot NO_3$, which are from

Gosse *et al.* (1997) for 0.7-2 µm and from Jarzembski *et al.* (2003) for 2-5.88 µm. As is the case for $NH_4 \cdot NO_3$, imaginary RIs

are set to an arbitrary small number ($1 \times 10^{-8}$) from 0.42-0.7 µm to reflect the little or no absorption in that spectrum (Palik and

Khanna, 1998). For the real RIs, in the 0.4-0.65 µm spectrum values are from Cotterell *et al.* (2017) for measurements at 0 %

relative humidity. From 0.66-0.67 µm, the real RIs are provided by Ballard *et al.* (1972) and for 0.7 µm from Ivlev and Popova

(1974). Above wavelengths of 1 mm - the scope of the Palik and Khanna (1998) database - the real and imaginary RIs are set

to the value at 1 mm. The compiled spectral RIs for $NH_4 \cdot NO_3$ and $NaNO_3$ are shown in Fig. S5 in the Supplement and tabulated

in Tables S3 and S4 in the Supplement.

The default configuration of UKCA-mode and by extension UKCA-Radaer, as used in UKESM1 and HadGEM3-GC3.1,

represents tropospheric $SO_4$ with $(NH_4)_2 \cdot SO_4$ refractive indices and (optionally) stratospheric $SO_4$ with sulphuric acid ($H_2SO_4$)

refractive indices. This is internally inconsistent given that the tropospheric $SO_4$ is missing the considerable mass associated

with $NH_4$. The new UKCA-mode nitrate configuration presented here that includes $SO_4$, $NO_3$, and $NH_4$ as separate tracers

firstly apportions $NH_4$ mass to $NO_3$ using a 1:1 molar ratio and the $NH_4$ remainder to $SO_4$ using a 2:1 molar ratio (represented

by $NH_4 \cdot NO_3$ and $(NH_4)_2 \cdot SO_4$ refractive indices respectively), and then represents remaining $SO_4$ mass with $H_2SO_4$ refractive

indices. Another internal inconsistency in the default UKCA-mode configuration is that $NH_4$ is not explicitly represented

during hygroscopic growth (i.e. in the ZSR algorithm), owing to the lack of an $NH_4$ tracer. In the new nitrate scheme, $NH_4$,

$NO_3$, and $hetNO_3$ are explicitly added to the hygroscopic growth routine, with $NH_4$ counteracting hygroscopic aerosol growth

and $NO_3$ and $hetNO_3$ promoting it.

**2.3 Simulation design**

The scientific purpose of this study is to investigate whether representing the kinetic limitation of $HNO_3$ condensation onto

pre-existing aerosols during the production of ammonium nitrate significantly alters the resulting atmospheric concentrations

of ammonium nitrate and, indirectly, coarse nitrate aerosol. To this end, four sensitivity simulations are performed with the

UM and the new nitrate scheme: 1. A control simulation with no nitrate aerosol and the default UKCA-mode setup 2 (i.e.

standard GA7.1) (CNTL); 2. A simulation with $NH_4 \cdot NO_3$ reaching equilibrium instantaneously (INSTANT); 3. A simulation

with the $HNO_3$ uptake rate set to $\gamma = 0.193$ in Eq. 7 following Feng and Penner (2007) (FAST); and 4. A simulation with the

$HNO_3$ uptake rate set to $\gamma = 0.001$ in Eq. 7 following Bauer *et al.* (2007) (SLOW). These simulations are further listed in

Table 2 and were selected to span the range of $HNO_3$ uptake rates on standard atmospheric particles from the literature (Bauer

*et al.*, 2007). All simulations are run for 25 model years with only the last 20 years used for analysis.




| Simulation name | Description |
|---|---|
| CNTL | Control simulation - no nitrate aerosols |
| INSTANT | Nitrate aerosols – instantaneous equilibrium for $NH_4NO_3$ ($\tau = 0$ in Eq. 8) |
| FAST | Nitrate aerosols– fast uptake coefficient for $NH_4NO_3$ ($\gamma = 0.193$ in Eq. 7) |
| SLOW | Nitrate aerosols– slow uptake coefficient for $NH_4NO_3$ ($\gamma = 0.001$ in Eq. 7) |

**Table 2: A description of the UM simulations performed in this study**

In these simulations, GA7.1 is forced by fixed sea surface temperature and sea ice fields prescribed as monthly climatologies for the year 2000, created by averaging over 1995-2004 the time-series data generated for CMIP6 atmosphere-only simulations. Additionally, aerosol and gaseous emissions are primarily prescribed as monthly fields from the CMIP6 historical emissions inventory (DECK/ Historical CMIP6 version 6.2.0), averaged over the 1995-2004 time-period. Table S2 in the Supplement gives global and annual total emissions for each of the UKCA chemical species. The CMIP6 emissions inventory was derived

from the Community Emissions Data System (CEDS) project which is documented by van Marle *et al.* (2017), Hoesly *et al.* (2018) and Feng *et al.* (2020); while its integration within the UM is detailed by Sellar *et al.* (2020). The simulation design (i.e. perpetual year 2000 conditions) follows standard simulation protocol for UKCA model development in the Met Office.

Global anthropogenic $NH_3$ emissions in the year 2000 from CEDS amount to 50 Tg yr$^{-1}$, in vast excess of equivalent emissions

from the CMIP5-derived MACCitty inventory of 37.5 Tg yr$^{-1}$ (Granier *et al.*, 2011). Hoesly *et al.* (2018) attribute this disparity to differing assumptions in agricultural $NH_3$ trends and to the lack of consideration for wastewater and human waste $NH_3$ emissions in MACCity. Oceanic $NH_3$ emissions in these simulations – which account for 26 % of total $NH_3$ emissions – follow Bouwman *et al.* (1997) and biomass-burning emissions are described by van Marle *et al.* (2017). $NH_3$ exhibits a strong seasonal cycle with global emissions in June-August ~50 % greater than in December-February (Fig. S6b in the Supplement). The

global $NH_3$ source of ~65 Tg yr$^{-1}$ in these simulations is close to the model-mean value of 63 Tg yr$^{-1}$ for GCMs participating in the AeroCom phase III nitrate experiment (Bian *et al.*, 2017). Nitrogen oxide ($NO_x$) emissions from anthropogenic, biomass-burning and aircraft sources are prescribed as monthly fields from the CEDS inventory (van Marle *et al.*, 2017; Hoesly *et al.*, 2018). $NO_x$ emissions from soils are taken from Yienger and Levy (1995), corrected to a total source of 12 Tg[NO] yr$^{-1}$ (Sellar *et al.*, 2020). The global and annual total $NO_x$ emissions amount to 106 Tg[NO] yr$^{-1}$. Further details on gas and aerosol

emissions in these simulations is provided in Section S3 in the Supplement.



## 3 Results

### 3.1 Global and annual mean metrics

Table 3 shows global tropospheric and annual mean budgets for $HNO_3$, $NH_3$, $NO_3$, and $NH_4$ from the FAST and SLOW simulations alongside equivalent metrics from the present day simulations of Xu and Penner (2012) (hereafter XP12),
Hauglustaine *et al*. (2014) (hereafter HA14), and from the AeroCom model intercomparison project detailed by Bian *et al*. (2017) (hereafter BI17). The INSTANT simulation is near-indistinguishable from FAST using these metrics (Table S5 in the Supplement) - suggesting that $NH_4 \cdot NO_3$ concentrations in FAST reach thermodynamic equilibrium near instantaneously - and INSTANT is thus omitted from further analysis. With respect to Table 3 and to the rest of the Results section, 'fine $NO_3$' refers to $NO_3$ associated with $NH_4$ while 'coarse $NO_3$' refers to $NO_3$ associated with dust and sea-salt (i.e. $NO_3$ in $hetNO_3$).


The net $HNO_3$ production rates in FAST (44.1 Tg[N] yr$^{-1}$) and SLOW (44.2 Tg[N] yr$^{-1}$) are comparable to equivalent rates in HA14 (45.1 Tg[N] yr$^{-1}$) and XP12 (38 Tg[N] yr$^{-1}$). Additionally, the $NH_3$ emissions in FAST and SLOW (53.5 Tg[N] yr$^{-1}$) are comparable to HA14, XP12, and BI14 (50.5, 53.6, and 51.8 Tg[N] yr$^{-1}$ respectively) suggesting that to the first order the ammonium and nitrate precursor gas emissions are commensurate on a global basis with prior studies. Total $NO_3$ (i.e. fine +
coarse) production in the FAST (22.9 Tg[N] yr$^{-1}$) and SLOW (19.9 Tg[N] yr$^{-1}$) simulations is significantly greater than in HA14 (14.4 Tg[N] yr$^{-1}$) and XP12 (16 Tg[N] yr$^{-1}$) and at the upper range of the AeroCom models in BI17 (Mean = 13.7, Range = 1.5 to 28.2 Tg[N] yr$^{-1}$). This is also the case for $NH_4$ production rates where FAST (30.4 Tg[N] yr$^{-1}$) and SLOW (25.6 Tg[N] yr$^{-1}$) exceed equivalent values in HA14 (17.5 Tg[N] yr$^{-1}$) and BI17 (Mean = 23.7, Range = 17.8 to 29.5 Tg[N] yr$^{-1}$), and are comparable with XP12 (30.5 Tg[N] yr$^{-1}$). This suggests that $NH_4$ and $NO_3$ aerosol production in the UM is at the upper end of
efficiency when compared to other existing climate models.

Significant differences between the FAST and SLOW simulations are highlighted by the global $NO_3$ metrics in Table 3. In particular, the fine $NO_3$ source is 6.3 Tg[N] yr$^{-1}$ in FAST but only 2.7 Tg[N] yr$^{-1}$ in SLOW, marking a 57 % decrease. Conversely, SLOW exhibits 5 % more coarse $NO_3$ production than in FAST. The difference is equally discernible in the
burdens, with 47 % of the total $NO_3$ burden as coarse $NO_3$ in FAST compared to 67 % in SLOW. This can be compared to a 72 % coarse fraction in HA14 and 47 % in XP12. The total $NO_3$ burdens of 0.2 Tg[N] in FAST and 0.15 Tg[N] in SLOW are commensurate with 0.18 Tg[N] in HA14, 0.17 Tg[N] in XP12, and the AeroCom median of 0.14 Tg[N] in BI17. The $NH_3$ burden in FAST (0.04 Tg[N]) is at the lower end of the AeroCom range in BI17 (0.04 to 0.7 Tg[N]) while the $NH_4$ burden in FAST (0.42 Tg[N]) is at the upper range of BI17 models (0.13 to 0.58 Tg[N]) which corroborates the assertion that $NH_4$ and
$NO_3$ aerosol production in the UM is at the upper end of efficiency when compared to other existing climate models, and suggests that $NH_3$ rather than $HNO_3$ is the limiting factor controlling $NH_4 \cdot NO_3$ production in these simulations. In summary, Table 3 illustrates the close parity with regards global and annual mean metrics between the UM simulations and previous nitrate simulations with various climate models from the literature.



|  |  |  |  | FAST | SLOW | XP12 | HA14 | BI17 |
|---|---|---|---|---|---|---|---|---|
| HNO$_3$ | Source | Gas phase | Tg[N] yr$^{-1}$ | 35.2 | 35.7 | 24.4 | 44.6 | [82, 92] |
|  |  | Aerosol phase |  | 18.6 | 18.2 | 17.9 | 3.9 | [4.7, 28.5] |
|  |  | Total |  | 53.8 | 53.9 | 42.3 | 48.5 | - |
|  | Loss | Gas phase | Tg[N] yr$^{-1}$ | 9.7 | 9.7 | 4.3 | 3.4 | [47, 66] |
|  |  | Fine nitrate |  | 6.4 | 2.7 | 8.8 | 3.2 | [2, 9.5] |
|  |  | Coarse nitrate |  | 16.6 | 17.3 | 7.2 | 11.2 | - |
|  |  | Dry deposition |  | 6 | 8 | 7.8 | 14.7 | 10.9 [8, 16.4] |
|  |  | Wet deposition |  | 14.9 | 15.8 | 14.5 | 17 | 25.1 [11, 37.2] |
|  |  | Total |  | 53.4 | 53.4 | 42.6 | 49.5 | - |
|  |  | Wet fraction | % | 71.4 | 66.4 | 65 | 53.6 | 68.6 [57.8, 76.3] |
|  | Burden |  | Tg[N] | 0.48 | 0.48 | 0.3 | 0.3 | 0.56 [0.15, 1.27] |
|  | Lifetime |  | days | 3.2 | 3.2 | 4.8 | 2.3 | [3.5, 5.7] |
| NO$_3$ | Source | Fine nitrate | Tg[N] yr$^{-1}$ | 6.3 | 2.7 | 8.8 | 3.2 | - |
|  |  | Coarse nitrate |  | 16.6 | 17.3 | 7.2 | 11.2 | - |
|  |  | Total |  | 22.9 | 19.9 | 16 | 14.4 | 13.7 [1.5, 28.2] |
|  | Loss | Dry deposition | Tg[N] yr$^{-1}$ | 8.9 | 8.3 | 4 | 1.7 | 4.7 [0.3, 10.8] |
|  |  | Wet deposition |  | 14.3 | 11.8 | 12 | 12.7 | 9.9 [1.2, 20.5] |
|  |  | Total |  | 23.2 | 20.1 | 16 | 14.4 | 13.7 [1.5, 28.3] |
|  |  | Wet fraction | % | 61.7 | 58.6 | 75 | 88.2 | 77 [56.3, 90.8] |
|  | Burden | Fine nitrate | Tg[N] | 0.11 | 0.05 | 0.09 | 0.05 | - |
|  |  | Coarse nitrate |  | 0.09 | 0.1 | 0.08 | 0.13 | - |
|  |  | Total |  | 0.2 | 0.15 | 0.17 | 0.18 | 0.14 [0.03, 0.42] |
|  | Lifetime | Fine nitrate | days | 6.2 | 7.2 | 3.7 | 5.6 | - |
|  |  | Coarse nitrate |  | 2 | 2 | 4 | 4.2 | - |
|  |  | Total |  | 3.2 | 2.7 | 3.9 | 4.6 | 5 [2, 7.8] |
| NH$_3$ | Source | Emissions | Tg[N] yr$^{-1}$ | 53.5 | 53.5 | 53.6 | 50.5 | 51.8 [46.9, 58.1] |
|  | Loss | Gas phase | Tg[N] yr$^{-1}$ | - | - | - | 0.6 | - |
|  |  | NH$_4$ formation |  | 30.4 | 25.6 | 30.5 | 17.5 | 26.4 [18.4, 34.6] |
|  |  | Dry deposition |  | 17.4 | 20.4 | 12.7 | 21.3 | 15.4 [10.4, 24.1] |
|  |  | Wet deposition |  | 5.7 | 7.5 | 9.6 | 11.1 | 11 [5.6, 15.3] |





| | | | | FAST | SLOW | Xu and Penner | HA14 | BI17 |
|---|---|---|---|---|---|---|---|---|
| | | Total | | 53.4 | 53.5 | 53.6 | 50.5 | 53.2 [49.8, 57.9] |
| | | Wet fraction | % | 24.6 | 27 | 43 | 34.3 | 40.7 [24.5, 58.1] |
| | Burden | | Tg[N] | 0.04 | 0.06 | 0.07 | 0.09 | 0.16 [0.04, 0.7] |
| | Lifetime | | days | 0.28 | 0.41 | 0.46 | 0.63 | 0.72 [0.29, 0.98] |
| $NH_4$ | Source | $NH_3$ conversion | Tg[N] yr$^{-1}$ | 30.4 | 25.6 | 30.5 | 17.5 | 23.7 [17.8, 29.5] |
| | Loss | Dry deposition | Tg[N] yr$^{-1}$ | 5.7 | 4.7 | 4.5 | 2.5 | 4.5 [1.3, 16.3] |
| | | Wet deposition | | 24.9 | 21 | 25.9 | 14.9 | 20.7 [5.6, 34.6] |
| | | Total | | 30.5 | 25.7 | 30.4 | 17.4 | 25.2 [17.7, 37.4] |
| | | Wet fraction | % | 81.4 | 81.8 | 85.2 | 85.6 | 81 [25.6, 94.7] |
| | Burden | | Tg[N] | 0.42 | 0.36 | 0.26 | 0.22 | 0.25 [0.13, 0.58] |
| | Lifetime | | days | 5 | 5 | 3.2 | 4.5 | 4.3 [1.9, 9.8] |

**Table 3: Global and annual-mean metrics for nitric acid (HNO$_3$), nitrate (NO$_3$), ammonia (NH$_3$) and ammonium (NH$_4$) in the FAST and SLOW simulations compared to Xu and Penner (2012), Hauglustaine *et al*. (2014) (HA14) and the AeroCom phase III model intercomparison project described in Bian *et al*. (2017) (BI17). Square brackets in the BI17 column denoted the AeroCom inter-model range.**

### 3.2 Annual mean burdens and concentrations

Figure 1 shows the annual-mean mass burdens of NH$_4$, NO$_3$ and hetNO$_3$ in the FAST and SLOW simulations. While units of mg[N]m$^{-2}$ are used in Fig. 1, units of 'mg of substance per m$^2$' are used in the following text for direct comparison with HA14. Fine NO$_3$ associated with NH$_4$ is concentrated over land regions, particularly in the Northern Hemisphere. The fine NO$_3$ burden averaged over European land is 1 mg[NO$_3$]m$^{-2}$ in SLOW and 3 mg[NO$_3$]m$^{-2}$ in FAST. The total NO$_3$ over Europe is 3 mg[NO$_3$]m$^{-2}$ in SLOW and 5 mg[NO$_3$]m$^{-3}$ in FAST, which is close to the simulated present day values of 3-4 mg[NO$_3$]m$^{-2}$ in HA14. Fine NO$_3$ peaks in Europe over the Mediterranean at 5 mg[NO$_3$]m$^{-2}$ in SLOW and 12 mg[NO$_3$]m$^{-2}$ in FAST. South Asia exhibits the greatest regional fine NO$_3$ burdens with 8 mg[NO$_3$]m$^{-2}$ in SLOW and 14 mg[NO$_3$]m$^{-2}$ in FAST. The total NO$_3$ burdens over South Asia of 11 mg[NO$_3$]m$^{-2}$ in SLOW and 15 mg[NO$_3$]m$^{-2}$ in FAST are somewhat greater than equivalent values in HA14 of 5-10 mg[NO$_3$]m$^{-2}$. Conversely, the total NO$_3$ concentrations in East Asia (China) are smaller in these simulations (5 mg[NO$_3$]m$^{-2}$ in SLOW and 9 mg[NO$_3$]m$^{-2}$ in FAST) than in HA14 (10-20 mg[NO$_3$]m$^{-2}$). Over Central North America, the total NO$_3$ burden is 2 mg[NO$_3$]m$^{-2}$ in SLOW and 5 mg[NO$_3$]m$^{-2}$ in FAST, which compares to 3-4 mg[NO$_3$]m$^{-2}$ in HA14. Reconciling all these inferences, FAST exhibits twice as much fine NO$_3$ burden on average as does SLOW. Whereas fine NO$_3$ burdens are concentrated over land, coarse NO$_3$ (i.e. hetNO$_3$) is more evenly spread over the Earth and prevalent over maritime areas where it forms on sea-salt and aged dust particles (Figs 1g,h). Over European land, coarse NO$_3$ constitutes 31 % of the total NO$_3$ burden in FAST and 63 % in SLOW. Equivalent figures for East Asia are 15 and 30 %; South Asia – 10



Figure 1: Annual-mean NH₄, NO₃, and hetNO₃ burdens from the SLOW and FAST simulations

and 23 %; and Eastern North America – 45 and 83 %. Therefore, the partitioning of NO₃ between the coarse and fine modes is highly sensitive to the uptake rate of HNO₃ on ambient aerosol.

Figure 2 shows maps of annual-mean near-surface concentrations of NH₄, NO₃ and hetNO₃ in the SLOW and FAST simulations. The spatial distributions of fine NO₃ are similar to those reported in BI17 and HA14 with peak concentrations over North America, Europe, South Asia, South East Asia, and East Asia land regions, coincident with the highest NH₃ and NOₓ emitting regions (Figure S6 in the Supplement). The average total NO₃ concentrations over Europe are 1.5 and 3.5 μg[NO₃]m⁻³ in SLOW and FAST respectively, which can be compared to 4-5 μg[NO₃]m⁻³ in HA14. In Central North America, total NO₃ concentrations amount to 1 and 3 μg[NO₃]m⁻³ on average in SLOW and FAST, with 50 % and 15 % contributions from coarse NO₃. The regional-mean total NO₃ concentrations in East Asia amount to 3.5 and 6.5 μg[NO₃]m⁻³ and in South





**Figure 2: Annual-mean NH₄, NO₃, and hetNO₃ near-surface concentrations from the SLOW and FAST simulations**

Asia amount to 5.5 and 7.5 µg[NO₃]m⁻³, in SLOW and FAST respectively. Total NO₃ differences between FAST and SLOW are driven by changes to the fine NO₃ concentrations (Figs 2d-f), with comparatively minimal changes to coarse NO₃ (Figs 2g-i).

Figure 3 shows the zonal mean, vertical distribution of NO₃, NH₄ and hetNO₃ aerosol in the FAST and SLOW simulations. NH₄ reaches a greater altitude than fine NO₃ owing to its long-lived association with SO₄ aerosol (Figs 3a-b). Due to the high solubility of NH₃ gas and thus swift wet removal from the atmosphere, free ammonia is mostly limited to the bottom 1 km of the atmosphere (Bellouin *et al*., 2011), which limits the vertical extent to which NH₄·NO₃ may form (Figs 3c-d). This is further corroborated by Figure S8 in the Supplement which shows the 'gas ratio', defined as $([NH_3] + [NH_4] - 2 \times [SO_4])$ divided

mage_ref id="1" />


**Figure 3: Annual and zonal-mean NH₄, NO₃, and hetNO₃ concentrations vs altitude from the SLOW and FAST simulations**

by $([HNO_3] + [NO_3])$, with values greater than 1 indicating that conditions are $HNO_3$-limited and less than 1 indicating conditions are $NH_3$-limited (Ansari and Pandis, 1998). It is clear from Fig. S8 that $NH_4 \cdot NO_3$ production is $HNO_3$- limited at





**Figure 4: Regional and monthly-mean NH₄, NO₃, and hetNO₃ near-surface concentration time-series for the SLOW and FAST simulations for 10 'Giorgi regions' [Giorgi, 2006] (land-only) representing high NO₃ concentration areas.**


the surface over land regions but that conditions are ubiquitously NH₃-limited above altitudes of 1000 m. While NaNO₃ and Ca(NO₃)₂ are not volatile like NH₄·NO₃, they are instead associated with coarse particles that are readily removed from the atmosphere by gravitational sedimentation and wet scavenging, and thus remain confined to the lowest 1 km of the atmosphere (Figs 3e-f).

**3.3 Regional surface concentrations**

Figure 4 shows the seasonal trends in NO₃, NH₄ and hetNO₃ near-surface concentrations averaged over land in 10 'Giorgi' regions (Giorgi, 2006), selected due to their high fine NO₃ concentrations (Fig. 2). For most of the regions (NEU, MED, CNA, EAS, WAF, SQF, and SEA), NH₄ and fine NO₃ trends in both the FAST and SLOW simulations are tightly coupled to trends



in regional NH$_3$ emissions (Figure S7 in the Supplement), which further corroborates the notion that NH$_4$'NO$_3$ formation may
be limited in these regions by available NH$_3$. NH$_4$'NO$_3$ concentrations in the CAS, SAS, and CSA regions may be more
dependent on seasonal meteorology than other regions, for instance SAS (i.e. South Asia) experiences a strong summer
monsoon which would enhance wet deposition of NH$_4$'NO$_3$ during summer and thus reduce concentrations. SAS also has
consistently elevated NH$_3$ concentrations throughout the year and is thus less sensitive to seasonal trends in NH$_3$ emissions
(Zhu *et al.*, 2015). In all regions, NH$_4$ and fine NO$_3$ concentrations exhibit a strong seasonal cycle in both SLOW and FAST
while the seasonal cycle in coarse NO$_3$ is less apparent. In the SLOW simulation, coarse NO$_3$ concentrations are of similar
magnitude to fine NO$_3$ concentrations in NEU, MED, CAS, CAN, CSA, and WAF on a regional-mean basis (Fig. 4).

Figures 5 and 6 show the near-surface concentrations of HNO$_3$, NH$_4$ and total NO$_3$ over the US (Fig. 5) and Europe (Fig. 6) in
the FAST and SLOW simulations compared to observations from the Clean Air Status and Trends Network (CASTNet,
www.epa.gov/castnet, last access: January 2021, Finkelstein *et al.* (2000)) for the US and the European Monitoring and
Evaluation Programme (EMEP, http://ebas.nilu.no/, last access: March 2021, Torseth *et al.* (2012)) for Europe. In both
networks the sites are located so as to represent the wider region. Data processing and site selection for the observations follows
the methodology described in Hardacre *et al.* (2021), who have compared SO$_2$ and SO$_4$ concentrations from UKESM
simulations with CASTNet and EMEP observations. CASTNet and EMEP data are averaged over the period 1994-2013 where
available. For CASTNet there are a total of 16 Western sites and 33 Eastern sites (49 in total) that meet data processing criteria
in this study (where the east-west delineation is described in Hardacre *et al.* (2021)). For EMEP there are 59 sites for HNO$_3$,
59 sites for NH$_4$ and 80 sites for NO$_3$ meeting data processing criteria over the 1994-2013 timeframe. For the scatterplots in
Figs 5 and 6, model output is linearly interpolated to measurement sites. It is important to note that the absolute magnitudes of
concentrations are not directly comparable between the simulations and observations given that the simulations assume
520    constant NO$_x$ and NH$_3$ emissions based on the year 2000 whereas NO$_x$ and NH$_3$ emissions in reality are transient. This becomes
apparent when comparing the network-mean concentrations in the simulations with the observations (Figure S9 in the
Supplement) where there is a clear negative trend in HNO$_3$, NH$_4$ and NO$_3$ concentrations in both CASTNet and EMEP
observations from 1994 to 2013.

525    Figure 5 shows that the spatial distributions of HNO$_3$, NO$_3$ and NH$_4$ over the US are similar in FAST and SLOW, with peak
HNO$_3$ concentrations in the east and midwestern states reflecting industrial NO$_x$ emissions, and peak NO$_3$ and NH$_4$ in the
midwestern and central states reflecting agricultural NH$_3$ emissions (Park *et al.*, 2004). The absolute magnitudes of NH$_4$ and
NO$_3$ concentrations are closer to CASTNet observations in SLOW (Fig. 5j) than in FAST (Fig. 5k), but the spatial correlation
coefficients are better in the eastern US in FAST ($R = 0.87$ for NH$_4$ and $R = 0.61$ for NO$_3$) than in SLOW ($R = 0.77$ for NH$_4$
530    and $R = 0.06$ for NO$_3$). This suggests that the positive NO$_3$ (and correspondingly NH$_4$) biases in FAST may emanate from a
surplus of HNO$_3$ in the model, given that HNO$_3$ is positively biased in the eastern US in both FAST and, to an even greater
extent, SLOW (Fig. 5d).



**Figure 5: Annual-mean HNO₃, NH₄ and total-NO₃ near-surface concentrations in the FAST and SLOW simulations over North America compared to CASTNet observations averaged over 1994-2013. In (b), (c), (f), (g), (j) and (k), coloured contours show simulated concentrations while overlayed filled circles represent CASTNet observations. In (a), (d), (e), (h), (i) and (l), 'R' is the Pearson's correlation coefficient and 'MnB' is the mean bias between simulated and observed concentrations**

Over Europe, $NO_3$ and $NH_4$ concentrations are closer to EMEP observations in the SLOW simulation than in the FAST simulation (e.g. smaller mean biases in Figs 6d,g). $NO_3$ concentrations in both FAST and SLOW peak in the Po Valley (North Italy) and Benelux Region (Belgium and the Netherlands), in anecdotal concordance with Druge *et al*. (2019) and references therein. The Po Valley peak in both the EMEP observations and simulations is due to the entrapment of industrial air pollution by regional geography. The observed $NH_4$ and $HNO_3$ peaks over the Czech Republic may be attributable to high agricultural $NH_3$ emissions pre-2004, with concomitant concentration declines owing to the Gothenburg protocol (Fortems-Cheiney *et al*., 2016; Giannakis *et al*., 2019). Neither of the observed $NH_4$ or $HNO_3$ concentration peaks in the Po Valley or the Czech



**Figure 6: Annual-mean HNO₃, NH₄ and total-NO₃ near-surface concentrations for the FAST and SLOW simulations over Europe compared to EMEP observations averaged over 1994-2013 where available. In (b), (c), (e), (f), (h) and (i), coloured contours show simulated concentrations while overlayed filled circles represent EMEP observations. In (a), (d), and (g), 'R' is the Pearson's correlation coefficient and 'MnB' is the mean bias between simulated and observed concentrations**





Republic are well captured by FAST or SLOW simulations which may be attributed to the coarse model resolution employed here (N96) and the close proximity of measurement sites to $NH_3$ sources.


Figure 7 shows the statistical distribution of 6 hourly total $NO_3$ and $NH_4$ concentrations from the FAST and SLOW simulations interpolated to two EMEP supersites in the UK – Auchencorth Moss and Chilbolton Observatory – and compared with observed diurnal concentrations from those sites (UK AIR, https://uk-air.defra.gov.uk/data/, last access: 29 January 2021). Both sites use MARGA instruments – a combination of wet rotating denuders for gas measurements, and stream jet aerosol

collectors for aerosol measurements – allowing for accurate partitioning between the aerosol and gas phases for volatile ammonium nitrate (Aas *et al*., 2012; Twigg *et al*., 2016). Only data recorded at the precise hours of 06:00, 12:00, 18:00, and 24:00 UTC that has passed the Department for Environment, Food & Rural Affairs' (DEFRA) quality control is utilised from the observations. Figure S10 in the Supplement shows the equivalent statistics for $HNO_3$ and $NH_3$ gases compared to observations. It is clear from Fig. 7 that $NH_4$ and $NO_3$ are significantly reduced in the SLOW simulation with respect to the

FAST simulation. It is also clear that the diurnal cycle, with $NH_4$ and $NO_3$ peaking at night (24Z or 00Z) and early morning (06Z) is both observed and skilfully simulated in both FAST and SLOW. Interestingly, the model shows similar biases to the Met Office's AQUM in: an over prediction of $NO_3$ during night-time in the summer; a slight under prediction of $NO_3$ in the winter; a large over prediction of $HNO_3$ during the day in the summer; and a smaller over-prediction of $HNO_3$ at night in the winter. The curious over-prediction of $NH_3$ at night in winter (Fig. S10 in the Supplement) was also observed in previous

incarnations of the AQUM and will be addressed in the UM in future by imposing a diurnal cycle to $NH_3$ emissions.

Figure 8 shows the seasonal cycle in total $PM_{2.5}$ concentrations in the CNTL, FAST, and SLOW simulations compared to collocated $PM_{2.5}$ observations from the Global Aerosol Synthesis and Science Project (GASSP; http://gassp.org.uk/data/, last access: 2 July 2020, Reddington *et al.*, 2017). GASSP amalgamates non-urban $PM_{2.5}$ measurements from three major networks:

The Interagency Monitoring of Protected Visual Environments (IMPROVE) project in North America, the European Monitoring and Evaluation Programme (EMEP), and Asia-Pacific Aerosol Database (A-PAD). The $PM_{2.5}$ analysis proceeds as in Turnock *et al.* (2020), with monthly-mean observations determined for each measurement site averaged over the years 2000-2010 and over each region. This is then compared with the simulated $PM_{2.5}$ output that has been linearly interpolated to individual site locations and averaged over the same time period and regions. Also shown in Fig. 8 is Modern-Era Retrospective

Analysis for Research and Applications, version 2 (MERRA-2) reanalysis data (Buchard *et al.*, 2017; Randles *et al.*, 2017), which closely follows GASSP surface observations in Europe and North America but is less successful in other regions where a smaller number of ground based observations are available, for example, incorrectly modelling the seasonal $PM_{2.5}$ cycles in South Asia and Pacific AUS NZ.





**Figure 7: December-February (DJF) and June-August (JJA) diurnal cycles of near-surface total-$NO_3$ and $NH_4$ aerosol concentrations in the FAST and SLOW simulations interpolated to two European Monitoring and Evaluation Programme (EMEP) supersites in the UK – Auchencorth Moss [55.79216ºN, -3.2429ºE] and Chilbolton [51.149617ºN, -1.438228ºE], alongside 6 hourly observations from 2014-2015 for Auchencorth Moss and 2016-2020 for Chilbolton**





**Figure 8: Regional and monthly-mean surface PM₂.₅ concentrations in the UM simulations (red, blue green lines), CMIP6 ensemble means, and from MERRA reanalyses and GASSP surface observations**

The largest observational networks in GASSP are over North America (IMPROVE) and Europe (EMEP). Over North America, the slight negative PM$_{2.5}$ bias in CNTL is brought closer to observations in SLOW and over-corrected in FAST which now exhibits a slight positive bias (Fig. 8). Over Europe observations suggest that PM$_{2.5}$ slightly peaks in DJF, which is not the case in any of the UM simulations in which PM$_{2.5}$ peaks in JJA. Druge *et al*. (2019) observed the same seasonal bias over Europe in the ALADIN-Climate regional model, which they attributed to uncertainties in the annual cycle of NH$_3$ and HNO$_3$ precursor gases. In Fig. 7, the simulated NH$_4$ concentrations in JJA vastly exceeded the observations at both UK-based supersites, indicating that summertime NH$_3$ gas emissions may be biased high in the prescribed CEDS emissions dataset over Europe. Over South Asia, East Asia, and South East Asia the CNTL simulation adeptly captures the seasonal PM$_{2.5}$ cycle and the addition of ammonium and nitrate in FAST and SLOW induces a slight positive bias. Figure 9 collates the annual mean





**Figure 9: Scatterplots of observed vs simulated annual-mean PM2.5 concentrations in the CNTL, SLOW and FAST simulations accompanied with relevant statistics – the normalised mean bias (NMB), the root mean square error (RMSE), the correlation coefficient ($r^2$) and the ratio between standard deviations (Sigma)**

observed and simulated $PM_{2.5}$ concentrations for each GASSP site, therefore statistics such as the normalised mean bias (NMB) are mostly weighted to the expansive North American and European networks. Nevertheless, it is interesting to note from Fig. 9 that the slight negative NMB in CNTL (-0.13) is completely compensated in SLOW (+0.02) and overcompensated in FAST (+0.26). However, all of the above inferences must be predicated on the fact the UM simulations use aerosol and gas emissions centred on the year 2000 whereas the observations are a composite of the years 2000-2010 and thus the temporal differences in emissions may partially explain disparities between the UM and observations. A robust result from Figs 8 and 9 is that both SLOW and FAST tend to increase $PM_{2.5}$ concentrations – particularly over Europe, Asia, and North America – with FAST inducing approximately double the $PM_{2.5}$ change than in SLOW.



**Figure 10: Annual-mean and column-integrated net-production and total deposition rates for NH₄, NO₃, hetNO₃, NH₃ and HNO₃, alongside the 'wet fraction' (the ratio of wet scavenging to total deposition), in the FAST simulation**



### 3.4 Ammonium and nitrate production and deposition

Figure 10 shows the column-integrated annual-mean net production (or emission) rate, total deposition rate, and wet deposition fraction for $NH_4$, $NO_3$, het$NO_3$, $NH_3$ and $HNO_3$ in the FAST simulation. $NH_4$ and fine $NO_3$ are primarily produced over land regions near $NH_3$ sources (Fig. 10j), with net $NO_3$ dissociation over adjacent land regions and oceans (Figs 10a,d). Figure S11 in the Supplement shows that whereas net $NH_4$ production is positive on a zonal-mean basis at altitudes up to 6 km, net fine $NO_3$ production is negative above 1 km north of 30°N latitude. Most fine $NO_3$ deposition is confined to land regions (Fig. 10e)

while coarse $NO_3$ is primarily deposited over oceans (Fig. 10h). Total deposition rates over land are 16, 5, and 4 Tg[N] $yr^{-1}$ for $NH_4$, fine $NO_3$ and coarse $NO_3$ respectively and 18 and 12 Tg[N] $yr^{-1}$ for $NH_3$ and $HNO_3$ gases respectively. Total deposition rates over oceans amount to 15, 2, and 13 Tg[N] $yr^{-1}$ for $NH_4$, fine $NO_3$ and coarse $NO_3$ respectively and 15 and 9 Tg[N] $yr^{-1}$ for $NH_3$ and $HNO_3$ gases respectively. This indicates that the dominant pathway for fixed nitrogen deposition to the oceans in these simulations, excluding river discharge sources, is via $NH_4$, $NH_3$ and coarse $NO_3$ deposition. The spatial distribution of

chemical production and deposition is similar in the FAST (Fig. 10) and SLOW (Fig. S12 in Supplement) simulations.

### 3.5 Aerosol optical depth and radiation changes

Figure 11 shows the annual-mean total aerosol optical depth at 550 nm ($AOD_{550}$) in the UM simulations (CNTL, FAST, and SLOW), with contributions from the Aitken, accumulation and coarse soluble modes, the Aitken insoluble mode, and mineral dust. Also plotted is the 2003-2012 mean MODIS 'collection 6' $AOD_{550}$ satellite data, which merges NASA's "Dark Target"

and "Deep Blue" algorithms and is widely used for validating aerosol models (Levy *et al*., 2013; Hsu *et al*., 2013). It is clear that generally the CNTL simulation does a reasonable job of simulating the spatial distribution of $AOD_{550}$ (Fig. 11a) when compared to MODIS (Fig. 11d).

The new nitrate scheme will impact the total $AOD_{550}$ by various direct and indirect routes. Firstly, the addition of $NO_3$, $NH_4$

and het$NO_3$ mass will increase the size and change the composition of the ambient aerosols thus altering their optical properties. Secondly, het$NO_3$ mass associated with sea-salt will replace existing NaCl thus changing the aerosol composition. Thirdly, the explicit addition of $NH_4$ to the hygroscopic growth routine will reduce hygroscopic growth, whereas $NO_3$ and het$NO_3$ will promote hygroscopic growth. Finally, tropospheric $SO_4$ was previously assumed to uniformly take the form of $(NH_4)_2SO_4$ in terms of optical properties, but in the new nitrate scheme is explicitly divided into $H_2SO_4$ and $(NH_4)_2SO_4$ contributions based

on $NH_4$ abundance. Indirectly, $NH_4$ and $NO_3$ may alter the $AOD_{550}$ by impacting online aerosol emissions (such as dust and sea-salt) and atmospheric oxidant concentrations in the UM.

The $AOD_{550}$ differences between the CNTL and nitrate simulations (FAST and SLOW) shown in Fig. 11 are a combination of the various direct and indirect changes to the UM listed above which do not necessarily result in an increase to $AOD_{550}$. The

global-mean $AOD_{550}$ difference between FAST and CNTL of +0.0048 is serendipitously close to equivalent nitrate AODs in





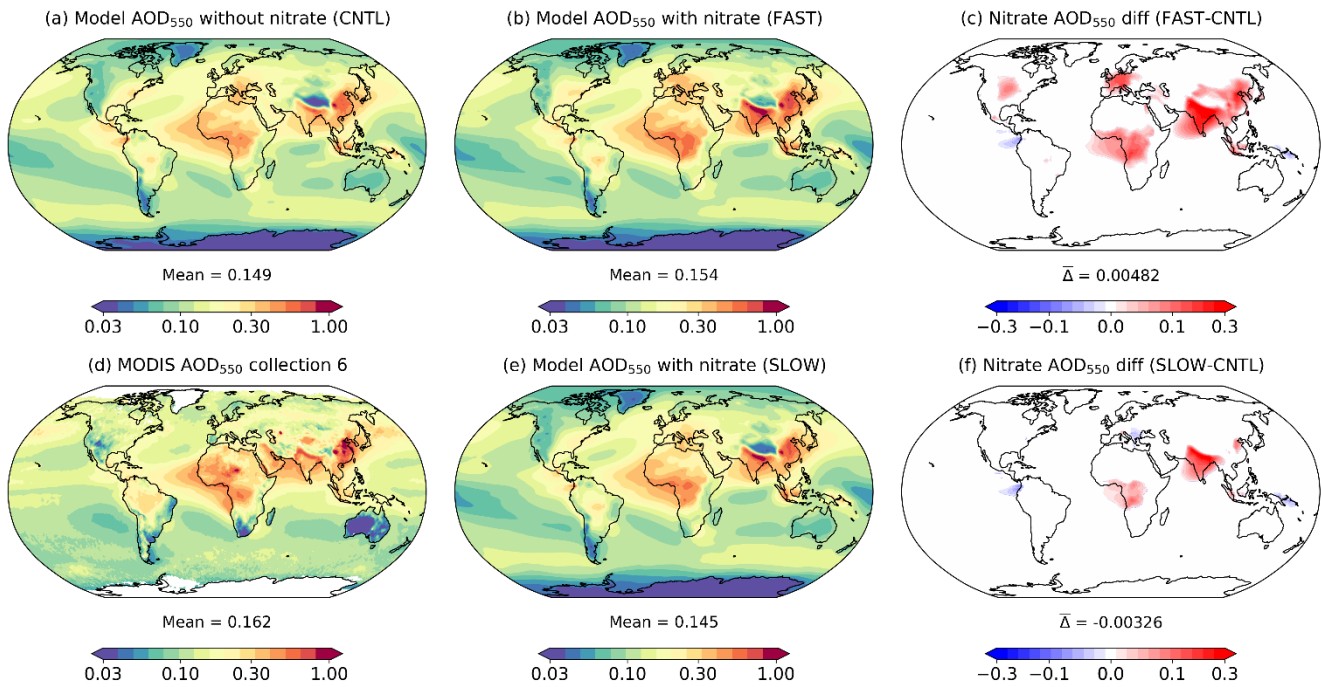

**Figure 11: Annual-mean and column-integrated 550nm aerosol optical depth (AOD$_{550}$) in the CNTL, FAST, and SLOW simulations and from MODIS collection 6 satellite observations. (c) and (f) show the difference between the FAST and CNTL, and SLOW and CNTL, simulations respectively**


the literature such as +0.006 in Paulot *et al*. (2016), +0.005 in HA14, and +0.006 [+0.002, +0.009] in Myhre *et al*. (2013), although it is important to note that the AOD$_{550}$ changes in those studies were derived from difference between present-day (PD) and pre-industrial (PI) simulations, rather than PD with and without nitrate. This is a subtle but important difference which reduces comparability given that the PI simulations included nitrate, albeit at much smaller concentrations than in PD

(Hauglustaine *et al.*, 2014). Despite the negligible global AOD$_{550}$ difference between FAST and CNTL, there are significant regional perturbations such as +0.05 over Northern Europe as a whole, +0.08 over East Asia, +0.19 over South Asia, and +0.04 over West Africa and Southern Equatorial Africa (Fig. 11c). The changes in the SLOW simulation are more subtle, for instance, AOD$_{550}$ changes by -0.007 over Northern Europe, +0.013 over East Asia, +0.1 over South Asia, and +0.018 over West Africa and Southern Equatorial Africa (Fig. 11f).


Figure 12 shows the annual-mean all-sky radiative flux perturbation at the Top-Of-the-Atmosphere (TOA). As the UM simulations are atmosphere-only with fixed sea-surface temperatures and sea-ice fields, the TOA radiative flux perturbation can also be denoted the total Effective Radiative Forcing (ERF) (Bellouin *et al.*, 2020). The global-mean ERFs from FAST-CNTL and SLOW-CNTL are -0.19 Wm$^{-2}$ and -0.07 Wm$^{-2}$ respectively, which is a similar magnitude to the PD-PI nitrate RFs



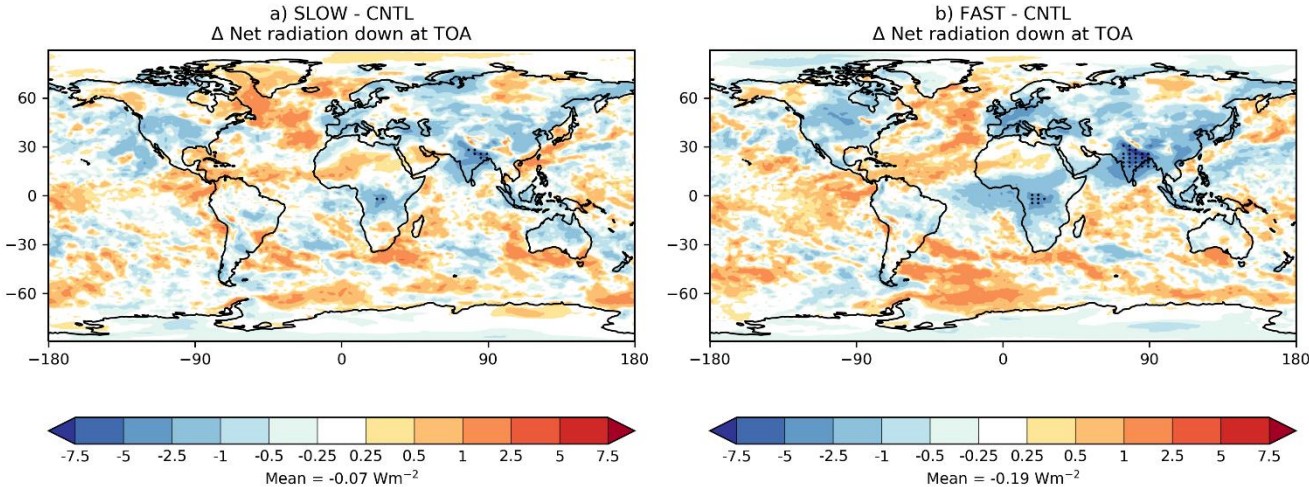

**Figure 12: Annual-mean Top-Of-the-Atmosphere (TOA) net downwelling radiative flux (shortwave + longwave) perturbation or effective total radiative forcing in the a) SLOW and b) FAST simulations with respect to CNTL. Stippling indicates where differences are significant at the ±2σ level**

of -0.056 Wm$^{-2}$ in HA14, -0.17 Wm$^{-2}$ in Bellouin *et al.* (2011), and -0.08 [-0.02, -0.12] Wm$^{-2}$ in Myhre *et al.* (2013), although note again that the RFs in those studies were derived from difference between present-day (PD) and pre-industrial (PI) simulations, rather than PD with and without nitrate, and those studies determined the direct radiative forcing rather than the total ERF which also includes indirect radiative impacts of cloud and atmospheric composition changes. An explicit assessment of nitrate-induced changes to cloud properties is outside the scope of this study. Although most regions exhibit insignificant radiative differences between FAST and CNTL in Fig. 12b (where stippling indicates significant changes at the ±2σ level), there are significant changes over Europe (min = -3.7 Wm$^{-2}$, mean = -1.5 Wm$^{-2}$), South Asia (min = -10.3 Wm$^{-2}$, mean = -3.2 Wm$^{-2}$), and Southern Equatorial Africa (min = -3.8 Wm$^{-2}$, mean = -1.2 Wm$^{-2}$). This mirrors nitrate's radiative signal in HA14. The SLOW simulation exhibits smaller radiative impacts than FAST, with significant changes limited to South Asia (min = -7.7 Wm$^{-2}$, mean = -1.5 Wm-2) and Southern Equatorial Africa (min = -3.5 Wm$^{-2}$, mean = -0.7 Wm$^{-2}$). Figure 12 highlights the regionality of ammonium nitrate climate forcing and further demonstrates the significant differences between the FAST and SLOW simulations.

## 4 Conclusions and Discussion

A thermodynamic equilibrium nitrate scheme has been added to UKCA-mode and tested in the Met Office's Unified Model. In contrast to widely utilised 'instantaneous' thermodynamic equilibrium models, the UKCA nitrate scheme limits the rate at which ammonium nitrate ($NH_4 \cdot NO_3$) concentrations reach equilibrium using first-order condensation theory. Sensitivity tests





are performed to assess the sensitivity of $NH_4.NO_3$ concentrations to the nitric acid ($HNO_3$) uptake coefficient ($\gamma$ in Eq. 7). Specifically, two values of $\gamma$ are chosen to represent fast uptake rates ($\gamma = 0.193$; FAST) and slow uptake rates ($\gamma = 0.001$; SLOW) based on the range of $\gamma$ measurements from the literature (e.g. Bauer *et al.*, 2004). While it is known that $\gamma$ varies with aerosol composition, temperature, and relativity humidity (e.g. Vlasenko *et al.*, 2006), well-constrained values for $HNO_3$

uptake on common aerosol species (e.g. sulphate, organic carbon, black carbon) are at present lacking. As a first order sensitivity test, the $HNO_3$ uptake coefficient used in $NH_4.NO_3$ production is assumed to be globally invariant to aerosol composition (with the exception of mineral dust), temperature and relative humidity in FAST and SLOW. A third nitrate simulation in which $NH_4.NO_3$ reaches thermodynamic equilibrium instantaneously (INSTANT), is shown to produce near-identical results to FAST.


To help evaluate the sensitivity of $NH_4.NO_3$ concentrations to $HNO_3$ uptake coefficient, and the suitability of the FAST and SLOW uptake coefficients, a range of surface and satellite observations and comparable modelling studies have been compared to the UM simulations. Many robust results emerge from the simulations. Fine $NO_3$ concentrations are a factor of 2 greater in FAST than in SLOW on a global-mean basis, with associated increases to $NH_4$ concentrations in FAST. The largest differences

are over land regions in North America, Europe, South and East Asia, and Equatorial Africa. However, there are minimal differences between coarse $NO_3$ (associated with dust and sea-salt) concentrations in FAST and SLOW. Over many populous land regions (Europe, North America, East and South East Asia, and West and Equatorial Africa), seasonal near-surface $NH_4.NO_3$ concentrations are closely correlated with seasonal $NH_3$ emissions suggesting that $NH_3$ availability is the limiting factor controlling $NH_4.NO_3$ prevalence (Giannakis *et al.*, 2019). In the SLOW simulation, coarse $NO_3$ concentrations are of a

similar magnitude to fine $NO_3$ concentrations over many industrialised regions. Comparing the simulated concentrations to CASTNet observations (i.e. the US network), FAST better captures the spatial distribution of near-surface $NO_3$, $NH_4$, and $HNO_3$ concentrations but is positively biased, whereas SLOW better captures the magnitude of the concentrations. Total $NO_3$ concentrations over Europe are comparable between SLOW and EMEP observations but are a factor of 3-4 too high in FAST. Many of the biases in simulated $NH_4$ and $NO_3$ concentrations appear to be artefacts of biases in precursor gas ($HNO_3$ and $NH_3$)

concentrations. Significant AOD and TOA radiative flux impacts are mostly isolated to land regions with substantial $NH_4.NO_3$ burdens. On a global mean basis, the nitrate ERF is -0.17 Wm$^{-2}$ in FAST and -0.07 Wm$^{-2}$ in SLOW which mirrors the ratio of $NH_4.NO_3$ burdens in the two simulations.

Introducing a kinetic limitation on the rate at which $NH_4.NO_3$ concentrations reach equilibrium has minimal effect for $\gamma = $

$0.193$ (i.e. comparing FAST with INSTANT) but a significant effect equivalent to a halving for $\gamma = 0.001$ (i.e. comparing SLOW with FAST). In general, FAST exhibits better spatial correlation with observed nitrate concentrations while SLOW better resolves the magnitude of concentrations. Note though that there are many caveats associated with this study. Using a globally uniform value for the $HNO_3$ uptake coefficient ($\gamma$) obviates the dependence of $\gamma$ on aerosol composition and relative humidity. A better parameterisation may instead utilise a volume-weighted $\gamma$ depending on aerosol composition and ambient



relative humidity. Additionally, assuming the same value of $\gamma$ for both $HNO_3$ and $NH_3$ is a pragmatic simplification owing to the dearth of $\gamma$ measurements. For example, Benduhn *et al*. (2016) assume uptake coefficients of 0.2 and 0.1 for $HNO_3$ and $NH_3$ respectively. On another note, if $\gamma$ is used to tune the $NH_4^.NO_3$ concentrations to observations in future then existing biases in precursor gases ($HNO_3$ and $NH_3$), in terms of emissions and atmospheric processes, should first be evaluated and addressed. For instance, the curious surplus of simulated $NH_3$ at night at UK sites (Figure S10 in the Supplement) may be

rectified by imposing a diurnal cycle on $NH_3$ emissions based on number of daylight hours, as implemented by Park *et al*. (2004). Bian *et al*. (2017) also highlight the importance of accurately simulating $NH_3$ dissolution in cloud droplets, which may be oversimplified in the UM owing to the ubiquitous assumption of a cloud droplet pH of 5 in UKCA. This study has also highlighted a potential overestimate of $NH_3$ emissions in Europe in the CMIP6 emissions inventory, as also posited by Druge *et al*. (2019). An accurate $NH_3$ and $NO_x$ emissions inventory is vital for a proficient simulation of $NH_4$ and $NO_3$ concentrations.

$HNO_3$ concentrations also appear to be overestimated over the western US (Fig. 5) in these simulations, which may emanate from an oversimplification of heterogeneous $N_2O_5$ chemistry in UKCA Strattrop1.0, given that the uptake coefficient in that reaction is uniformly set to 0.1 (Archibald *et al*., 2020).

The differences between the simulated and observed concentrations in this study (Figs 5-7) may be attributed to the use of

perpetual year-2000 conditions in these simulations; the coarse model resolution utilised here (N96); biases in $HNO_3$ and $NH_3$ emissions; deficiencies in the thermodynamic equilibrium approach; and due to the choice of a monotonic uptake coefficient. In particular, this study has shown that the $HNO_3$ uptake coefficient is an important parameter in the production of ammonium nitrate and assuming a monotonic value in climate models may be too simplified an approach given the high sensitivity of $HNO_3$ uptake to ambient aerosol composition. Future simulations would benefit from stronger observational constraints on the

$HNO_3$ and $NH_3$ uptake rates as a function of aerosol composition, relative humidity, and temperature, perhaps from targeted laboratory studies.

In a follow-on study, we aim to evaluate the nitrate scheme in high-resolution UM simulations for specific meteorological case-studies in a manner analogous to Gordon *et al*. (2018) but over a UK-based domain. Additionally, we will replace the

constant $HNO_3$ uptake coefficient in the new nitrate scheme with a volume-weighted value based on aerosol composition and relative humidity, and re-run the simulations using transient CMIP6-like atmosphere-only UM integrations. The next issue to address will be coupling $NO_3$ and $NH_4$ aerosol within the UKESM framework. At present, fixed nitrogen ($NO_y + NH_x$) deposition to the land-surface model (JULES) in UKESM is applied using offline deposition fields from the input4MIPs database (see Sellar *et al.* (2020) for further details). Meanwhile, the ocean biogeochemistry module in UKESM (MEDUSA2;

Yool *et al.* (2003)) has a closed nitrogen budget thus obviating interactions with atmospheric nitrogen. With the addition of ammonium and nitrate aerosol to a future version of UKESM, we will aim to fully couple atmospheric fixed-nitrogen deposition with the land and ocean surfaces to permit a comprehensive closed-budget nitrogen cycle.



In conclusion, the addition of ammonium and nitrate aerosol to UKCA-mode in the UM is step-change in aerosol-modelling
capability in the UK and will increase confidence in future simulations of aerosol forcing and regional air pollution episodes.
Additionally, nitrate concentrations have been shown to be highly sensitive to the nitric acid uptake rate, paving a way for
climate models to reduce outstanding biases in ammonium nitrate concentrations.

**Code Availability**

Due to intellectual property rights restrictions, we cannot provide either the source code or documentation papers for the UM.
The Met Office Unified Model is available for use under licence. A number of research organisations and national
meteorological services use the UM in collaboration with the Met Office to undertake basic atmospheric process research,
produce forecasts, develop the UM code, and build and evaluate Earth system models. For further information on how to apply
for a licence, see http://www.metoffice.gov.uk/research/modelling-systems/unified-model (last access: 16 April 2021). The
nitrate scheme is now available on the 'trunk' (the Met Office's data repository) and is available for all future UM versions
since vn11.8 in UKCA mode setup 10.

**Data Availability**

The UM data used to produce the figures is available from the Centre of Environmental Data Analysis (CEDA)
(http://dx.doi.org/10.5285/0613b74ecc574fa7b6ac8a22838c5f81; last accessed 12/05/2021) and if used should be cited
accordingly: Jones, A.C.; Hill, A.; Remy, S.; Abraham, N.L.; Dalvi, M.; Hardacre, C.; Hewitt, A.J.; Johnson, B.; Mulcahy,
J.P.; Turnock, S. (2021): Exploring the sensitivity of atmospheric nitrate concentrations to nitric acid uptake rate using the Met
Office's Unified Model: nitrate and nitric acid simulation data. NERC EDS Centre for Environmental Data Analysis, *date of
citation*. https://catalogue.ceda.ac.uk/uuid/0613b74ecc574fa7b6ac8a22838c5f81

**Competing Interests**

The authors declare that they have no conflict of interest.

**Author contributions**

ACJ developed the nitrate scheme with assistance from AH, SR, NLA, MD, and AJH. ACJ performed the simulations with
assistance from NLA and MD. ACJ, AH, CH, BJ, JM, and SR analysed the simulations and compared the results to
observations. ACJ wrote the manuscript with assistance from all co-authors.



**Acknowledgements**

The authors would like to thank the Met Office and the UM team for providing the UM climate model; NASA for making the MODIS satellite data freely available; and EMEP, the United States Environment Protection Agency (EPA), and GASSP for making their concentration measurement data freely available. The authors thank the many scientists who have contributed data utilised in this evaluation. The authors would also like to thank Didier Hauglustaine, John Hemmings, Nicolas Bellouin, Alexander Archibald, Paul Griffiths, Steve Rumbold, Michael Cotterell, and Paul Agnew for their

comments and assistance throughout this project. Figures were produced using Python 3.6.10 (https://www.python.org/) and Iris 2.4.0 (https://scitools.org.uk/).

ACJ and AH are supported by the Clean Air programme which is jointly delivered by the Natural Environment Research Council (NERC) and the Met Office, with the Economic and Social Research Council (ESRC), Engineering and Physical Sciences Research Council (EPSRC), Innovate UK, Medical Research Council (MRC), National Physical Laboratory (NPL),

Science and Technology Facilities Research Council (STFC), Department for Environment, Food and Rural Affairs (Defra), Department for Health and Social Care (DHSC), Department for Transport (DfT), Scottish Government and Welsh Government. AJH is supported by the Joint DECC/Defra Met Office Hadley Centre Climate Programme (GA01101). BJ is funded by the Met Office Hadley Centre Climate Programme funded by BEIS and Defra (GA01101). NLA is supported by NERC and NCAS through the ACSIS project.

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
