# Peer review of "Exploring the sensitivity of atmospheric nitrate concentrations to nitric acid uptake rate using the Met Office's Unified Model"

_Atmospheric Chemistry and Physics, 2021_

## Referee Comment (RC1)

**Overall Summary:**

This work describes a new thermodynamic equilibrium scheme developed, which explicitly considers kinetic limitations of $HNO_3$ and $NH_3$ uptake to aerosols in the UK's climate model. They test the common assumption in global models that $NH_4NO_3$ concentrations reach thermodynamic equilibrium instantaneously, by varying the assumed the heterogenous uptake rate of $HNO_3$ and $NH_3$ in the model and investigating the impacts on important climate parameters such as $PM_{2.5}$ concentrations, AOD, radiative balance, etc. They demonstrate that such a variation does have significant air quality and climate implications, and that the computational cost of implementing such a scheme is worthwhile. But, perhaps most significantly, they highlight the high sensitivity of ammonium nitrate concentrations to nitric acid uptake rates and, therefore, provide a mechanism for reducing nitrate concentrations biases in other climate models, which make similar instantaneous thermodynamic equilibrium assumptions.

Given the robustness of this problem in climate models beyond just this specific model, this paper represents a substantial contribution to scientific progress well within the scope of Atmospheric Chemistry and Physics. In this paper, the methods are clearly outlined, the conclusions reached are well supported, the abstract is clear, and importantly, even though the source code for the model cannot be provided due to licensing, the details in the paper and supplement are enough that the methods outlined could be reproduced in other models. The citations are appropriate and worth commending the authors for their efforts as it is obvious, they have a detailed understanding of the body of literature on this topic.  The title is appropriately descriptive, and overall, the paper is very readable. I did find the number of figures to be overwhelming and excessive in some places and have suggested a few ways to make the results section slightly clearer/ potentially shorter, which I believe would increase the reach and impact of the paper. However, overall, I recommend that the paper be accepted with minor revisions.

**General Comments:**

The abstract, introduction, background, motivation, methodology, and conclusions of the paper are particularly well written - perhaps one of the best of papers I have reviewed.  However, I found the results section of the paper overwhelming, at times, given the number of figures, the number of panels in figures, and the amount of information contained in any one figure. In stark contrast to the readability of the rest of the paper, I felt that the results section could be significantly improved by (1) the addition of a single summary sentence at the end of many paragraphs in order to better highlight conclusion / context of the conclusion that is reached and (2) by tightening up the figures (e.g. reducing the total number of figures, reducing the number of panels, or reducing the number of lines on any one figure). To be clear, I found the conclusions and abstract to be very understandable and well supported. However, I thought the communication of the points within the results section itself- particularly how the text discussion of figures is structured- could be improved. In the specific comments I've tried to point out both where this is done well and where, even as a "target reader" I initially struggled to follow.

One question I did walk away from the paper with, is whether the testing of the model's sensitivity to instantaneous equilibrium was truly novel? I am curious as to whether this is the first paper exploring this or whether it is something that was implemented in the UM following the implementation of something similar in a different model. I'm not familiar enough with what's done in models beyond GEOS-Chem (which uses ISORROPIA II & therefore the instantaneous equilibrium assumption), but it strikes me that how novel this approach is this should be made more clear in the final paper- either in order to point out the truly novel result of this work, or to acknowledge that which came before in a way that is more explicit than is done now.

As a small final note, I found it generally confusing that the authors refer to coarse mode nitrate as hetNO3. The terms are used interchangeably in the main text and discussion, but I think just changing the moniker hetNO3 to something like $coarseNO_3$, $cNO_3$, $bigNO_3$ would be less confusing. Simply CNTRL+F'ing the text to ensure consistency in their reference to this would certainly improve the communication of the result.

**Specific Comments:**

**Line 123:** …"model is yet to successfully"- Typo: I think "yet" should be "set"

**Line 176-177:** I highlighted this line because on my initial read through, I was concerned that the assumption of a static cloud pH could impact the results contained within. I saw much later in the discussion that this assumption was addressed- it may be worth mentioning that you address this later in Section XX.

**Lines 230-235 and 285-296:** I found myself struggling here to translate some of the mathematical statements to readable words. Instead of stating "Ta*TN > $K_p$" or  "Ta*TN < $K_p$" , I suggest the authors put what this means into physical context using words for the readers. For example: "If available nitrate & ammonium suggest the equilibrium of Reaction 2 is in the forwards direction, thereby promoting the condensation of $HNO_3$ and $NH_3$ to form $NH_4NO_3$ (Ta*TN > $K_p$), then the equilibrium concentration of $NH_4NO_3$ it is solved using Eq 3. Otherwise…"

**Lines 293-295:** I was confused about the assumption that ammonium nitrate does not nucleate new particles and thought some elaboration was needed here. There's a recent Nature paper showing that ammonium nitrate out of equilibrium in clouds can be important nucleation/growth events even if such a state is only reached for a few minutes (https://www.nature.com/articles/s41586-020-2270-4). This is likely less important within a global climate model, but it is worth at least a sentence or two of elaboration.

**Line 309:** Why is alkalinity not titrated for sea-salt? Is it just an assumption that $HNO_3$ is in excess of $Na^+$? I found it odd that this was stated with no reasoning.

**Line 389:** It seems to me that the uncertainty in emissions driving the model are a major contributor to the discrepancies between the model and observation comparisons made in the results section- which is why there is time spent making seasonal and spatial comparisons between the model and observation networks. I think its worth a sentence here preparing the reader for that pointing out the significant of these emissions uncertainties- something along the lines of: "Such large discrepancies in $NH_3$ emissions

inventories can impact direct model- measurement comparisons … which make it important to consider the spatial and temporal trends rather than just the overall magnitudes. For this work exploring the sensitivity of $NH_4NO_3$ to thermodynamic equilibrium assumptions, the direct comparison of model performance to observations is done with the goal of understanding the degree to which thermodynamic assumptions may push the model out of realistic behavior rather than best re-creating observations"…

**Line 419-420:** I wanted to highlight these lines, because I they do a really good job of "summarizing" what the previous paragraph states / contextualizing the results pointed out. It is what I was longing for in other parts of the results section. I suggest the authors look through the other final sentences in the results section and model them after this paragraph.

**Lines 421-434:** This is one of the results paragraphs that is not well contextualized to the physical chemistry occurring/ the overall point of the results communicated. The authors point out the differences in the fast and slow fine and coarse model $NO_3$ sources and burdens, coarse fractions & how they compare to other models. But they leap to " these results… corroborates the assertion that NH4 and $NO_3$ aerosol production is at the upper end of efficiency & that $NH_3$ is limiting" without clearly explaining why. (e.g. Fine mode production of $NO_3$ is faster in FAST (ok, duh), but coarse mode production of NO3 is faster in SLOW (why physically? Is it that there is more $HNO_3$ in the gas phase because $NH_4NO_3$ production is slower, so that more $HNO_3$ is available to condense on the coarse mode particles?) Then contextualize WHY that the % of the $NO_3$ burden in the coarse mode of SLOW is closer in HA14 and FAST is closer to XP12 – what assumptions do those studies make about the $HNO_3$ uptake (instantaneous?). Is the % of $NO_3$ burden in coarse mode comparisons just to say that it's within the range of other models? That's what the final sentence seems to suggest. This just needs to be more clear- there's a sentence missing connecting the stated results to the conclusion made.

**460:** Nicely communicated & contextualized result.

**Figure 4:** This figure was particularly hard for me to decipher, and I spent far too long trying to understand the message it was communicating. The conclusion in the text is the only reason I was able to follow what this was showing. First, the key has cut off the symbol for "hetNO$_3$ Fast" so that its symbol looks identical to $NO_3$ fast- I assume that the solid line is $NO_3$ fast and the long dashes are hetNO$_3$, but the key does not distinguish them.  I think this plot could be greatly improved if it was more clear which lines we're supposed to be comparing on each figure. Right now, that is done with the different line styles (e.g. compare black solid like to black solid line), but I think the line thickness are different between the fast and slow simulations, which makes identifying which lines are supposed to be comparable difficult (e.g. the dashed black line looks totally different than the color dashed line). Another way to improve this figure would be to improve the key- Put the slow and fast lines that are comparable next to one another (so 2 columns in the key).

 **Line 499-506:** Clarity of result could be improved/ needs a better summary sentence.

**Line 518:** The authors mention several times that the "model output is linearly interpolated to measurement sites", but its not totally clear exactly what that means. Is it that value assumed at the specific lat/long of the site is actually interpolated from its distance to other grid boxes & their values?

**Line 525-533:** This is another section where I was looking for a better summary sentence. I'm not sure if the point of these figures is to show that FAST/SLOW is "better" at simulating observations or that simply changing the assumption about thermodynamic equilibrium has a huge impact on $NO_3$, $NH_4$, and $HNO_3$. Figure 5 and 6 are pushing me to decide which assumption is "better"- but the emissions bias seems to muddle making a direct conclusion. E.g. SLOW looks closer to the 1:1 line, but it's only there because of emissions biases? Does this mean FAST is better? Is that the point of the figure? Perhaps you could add a "theoretical" 1:1 line to show what you'd expect if there weren't emissions biases? The conclusions section describing this result is much clearer (lines 710-715) than the results discussion here.

**Line 597-599:** I was also wondering about $N_2O_5$ chemistry being promoted in the winter & how that is simulated in the model? Perhaps worth a mention here as a reason for the PM2.5 disagreements. (Maybe see Shah et al., 2018:  https://www.pnas.org/content/115/32/8110/tab-article-info).

**Figure 10 & Section 3.4**: I'm not sure that Figure 10 adds anything to the paper that cannot be described in the text. The figure has so many panels that it's very confusing to read and hard to compare the panels needed to get to the in-text conclusions. I suggest the authors consider moving it to the supplement or simplifying the figure.

**Figure 12:** Given the small region where the TOA radiation differences are significant between the two results, I also suggest cutting this figure or moving it to the supplement. The results can be adequately described in the text without a figure.

---

## Referee Comment (RC2)

**Review of manuscript titled "Exploring the sensitivity of atmospheric nitrate concentrations to nitric acid uptake rate using the Met Office's Unified Model" by Jones et al. submitted to Atmospheric Chemistry and Physics Discussions**

This work presents the UK Met office's Unified Model (UM) with a new nitrate scheme in which sensitivity of thermodynamic equilibrium for NH4NO3 production to the HNO3 and NH3 uptake to aerosol, is explored. Furthermore, the impact of these sensitivities on mass budget and total deposition rates for NH4, NO3 (fine and coarse), NH3 and HNO3, PM2.5, AOD and radiation changes etc. have been descriptively presented on a global scale. This work specifically provides a way to address overestimation of the NO3 aerosol by current crop of climate models, in a computationally efficient way.

In general, the manuscript is quite descriptive in terms of presenting the methods with governing equations (including supplementary material), which is valuable for reproducibility, and provides descriptive robust analysis and validation of its findings with observation networks across the globe (temporal and spatial comparisons), and previously reported aerosol budgets and lifetime. The work presented in this manuscript is no doubt quite robust and essential to pave way for more complex parametrizations to be added in future to UM. The introduction of this manuscript specifically can be referred to students of atmospheric chemistry as a broad overview of secondary inorganic aerosol formation and the state of their modeling (specifically NO3 aerosols).

**General Comment:**

The main overarching issues were pertaining to some detailed explanations in methods and their explanatory text in main manuscript that can either be summarized, tabulated and/or shifted to supplement to allow reader(s) to get at the results and discussions quickly, and not be overwhelmed before they get to that part. The lack of brevity specifically in 'Methods' and to some extent in 'Results' needs to be addressed by the authors before a revised version of manuscript can be accepted finally for publication (some suggestions in Specific Comments). Also, the distinction between fine vs coarse NO3 in terms of nomenclature say coarse mode nitrate as hetNO3 can be confusing to readers (See Specific Comments on Table 1). However, I encourage this manuscript to be accepted for publication in Atmospheric Chemistry and Physics journal with minor revisions.

**Specific Comments:**

**Lines 40-60:** Make sure the paragraphs are Justified instead of left aligned and ensure the same in rest of manuscript consistently.

Lines 122-123: Edit the following as:

"The hybrid-dynamical nitrate scheme developed by Benduhn et al. (2016) in the standalone GLOMAP-mode model is currently not implemented <del>yet to successfully transition to in the UM."</del>

Lines 125-126: Edit the following as: (if this is accurate)

**"...in order to fill the NH4 and NO3 shaped void. address the gaps in modeled $NH_4$ and $NO_3$ with their respective observations."**

**Section 2.1:** Will suggest the authors to consider summarizing the configuration of the UM used to test the new nitrate scheme as described in section 2.1 in tabular form either in main manuscript or even move to supplement. This will help readers to get to the focus of reader(s) quickly to the new nitrate thermodynamic scheme in UM quickly.

**Table 1:**

- a) please denote that  $\sigma g$  is referred to geometric standard deviation in the Table 1 caption.
- **b)** Can the authors enunciate more on the difference between NO3 and hetNO3 in accumulation and coarse mode? (Lines 209-210 ["*NH4 and NO3 mass is emitted into the Aitken and accumulation soluble modes and may be transferred to the coarse soluble mode via aerosol processing, while hetNO3 is limited to the accumulation and coarse soluble modes."*] seems to explain this? But might help to etch it out more in terms of this question and Table 1).
- c) In Lines **409-410**: authors state "'fine NO3' refers to NO3 associated with NH4 while 'coarse NO3' refers to NO3 associated with dust and sea-salt (i.e. NO3 in hetNO3)." Similar clarification to address comment **b**) early on in the text would be helpful.

**Section 2.2.1:** Please ensure chronological ordering of supplemental figures (for instance S2 comes after S3 in text now), re-arrange in the order they are mentioned in text. Make sure Tables, figures are arranged in a chronological order in general throughout the manuscript.

**Lines 279-281:** *"However, atmospheric Aitken mode number concentrations generally exceed accumulation mode concentrations, particularly over populous land regions and increasingly with altitude."*

Referring to the above statement, did the authors observe any converse trend (i.e. higher accumulation ode concentrations) over say rural or coastal non-urban regions? Discussing the spatial patterns in Fig S4 briefly in 1-2 sentence(s) would be a valuable inference to add from the manuscript's findings.

**Lines 294-295:** Apart from Wang et al., 2020 (Nature) showing  $NH_4NO_3$  nucleating for new particle formation (NPF), there has been an increasing interest in exploring NPF parametrizations specifically for bridging the model-observation number concentration gaps at high elevations (https://agupubs.onlinelibrary.wiley.com/doi/10.1029/2018JD029356). It would be good to point this out may be in discussions on future steps (in conclusions), as NPF parametrizations are still an evolving area when it comes to global climate models.

**Lines 307-309 (see next comment as well on Section 4):** Further/clearer explanations on the merit of following assumptions would be better:

"Additionally, dust is assumed to uniformly constitute 5 %  $Ca^{2+}$  by mass, which differs from the approach in Remy et al. (2019) who used a spatially hetereogeneous  $Ca^{2+}$  fraction. Dust alkalinity is titrated by uptake of HNO3 until the dust pH is neutralised whereupon HNO3 stops condensing, while no such limitation is applied for sea-salt."

**Lines 390-400 (and Section 4):** Follow up in conclusion on how uncertainty in different emission sources specifically NH3 with overestimated inventory can be a major source of uncertainty. I see points in Section 4 has been made regarding the NH3 and NOx inventory-induced uncertainty and N2O5 chemistry simplification (Lines 735:738: "An accurate NH3 and NOx emissions inventory is vital for a proficient simulation of NH4 and NO3 concentrations. HNO3 concentrations also appear to be overestimated over the western US (Fig. 5) in these simulations, which may emanate from an oversimplification of heterogeneous N2O5 chemistry in UKCA Strattrop1.0, given that the uptake coefficient in that reaction is uniformly set to 0.1 (Archibald et al., 2020)."]. But similar impacts from say assumptions made in Lines 307-309 (for instance) can be further dissected in conclusions as they are missing.

**Figure 3:** The only fundamental conceptual critique I have on result section is pertaining to Fig. 3: is that the manuscript does not elucidate much on the 'role of Convection' in their version of UM model that can also bring nucleation precursors (NH4, NO3 for instance) from the ground level to the free troposphere? The vertical profile obviously is limited to 0-6 km as NPF or nucleation has not been included (see comment for **Lines 294-295**). This limitation is essential to be mentioned as stated earlier, albeit as a future step if beyond the scope of current manuscript.

**Figs 5,6 and 8:** The discussions around these figures seem to give similar inferences and can be synthesized together in terms of their summary pointing to similar inferences on FAST introducing positive biases etc. Its understandable than Figs. 5, 6 are annual means and Fig. 8 gives a seasonal variation, but still can be very much summarized together. Or some parts of them can be moved to supplement.

**Figure 11:** Authors can do away with Panels (a),(b), and (e) and their discussions on AOD results for CNTL, FAST and SLOW sensitivity runs, and just show difference maps (c, f). Can move the Panels (a),(b),(d),(e) as separate figure into supplement and summarize the model vs observed AOD in result text, referring to the new supplemental figure. There is a need to synthesize the result section, in terms of what take-out messages the authors want to stand out to the reader(s).

---

## Author Response (AR1)

**Response to reviewers for paper: "Exploring the sensitivity of atmospheric nitrate concentrations to nitric acid uptake rate using the Met Office's Unified Model" by A C Jones et al.**

We thank the Reviewers for their very useful comments that have significantly improved the manuscript. We are glad that the Reviewers see the merits of the study and we have endeavoured to address all of the suggestions they have made below. In particular, all of the extraneous detail in the Methods section has been moved to the Supplement and the Results section is much refined. Below, we have listed each of the Reviewers comments in red, replies in black, and highlighted changes to the manuscript in blue.

**Reviewer 1**

**General Comments**

> I felt that the results section could be significantly improved by (1) the addition of a single summary sentence at the end of many paragraphs in order to better highlight conclusion / context of the conclusion that is reached and (2) by tightening up the figures (e.g. reducing the total number of figures, reducing the number of panels, or reducing the number of lines on any one figure).

We agree with the Reviewer that the results section needed refining. We have therefore added many introductory remarks and concluding comments to the results section to improve its readability, as suggested. Additionally, we have simplified 2 figures (Figs 4 and 5) and removed 3 extraneous figures (Figs 9, 10, and 12) from the manuscript. We have also removed Section 3.4 from the manuscript as it contributed little. In our, opinion these refinements have greatly improved the manuscript's readability and scientific reach.

> One question I did walk away from the paper with, is whether the testing of the sensitivity to instantaneous equilibrium was truly novel? I am curious as to whether this is the first paper exploring this or whether it is something that was implemented in the UM following the implementation of something similar in a different model

To the author's best knowledge, this sensitivity test is truly novel. The sensitivity to the uptake coefficient has been explored for coarse nitrate (Bauer et al., 2004) but not for ammonium nitrate (or fine nitrate). Remy et al. (2019) introduced a time constraint to the thermodynamic equilibrium scheme which forms the basis of out model but used a fixed value of $\tau = 2$ mins, as explained in section 2.1. This manuscript is the first to relate the exponential decay time $\tau$ to properties of the underlying aerosol and temperature in a GCM, as recommended by Wexler and Seinfeld (1990). The following has been added to the end of Section 1 and to the Results section:

"This is the first study to investigate the sensitivity of $NH_4 \cdot NO_3$ concentrations to the $HNO_3$ uptake coefficient and provide a computationally efficient method for reducing $NO_3$ concentration biases in GCMs."

> As a small final note, I found it generally confusing that the authors refer to coarse mode nitrate as hetNO3. The terms are used interchangeably in the main text and discussion, but I

think just changing the moniker hetNO3 to something like coarseNO$_3$, cNO$_3$, bigNO$_3$ would be less confusing.

We thank the Reviewer for the suggestion. hetNO$_3$ has been changed to coarseNO$_3$.

**Specific Comments**

➢ Line 123: "Model is yet so successfully" – Typo: I think "yet" should be "set"

Following Reviewer 2's advice, this sentence has been changed to:

"The hybrid-dynamical nitrate scheme developed by Benduhn et al. (2016) in the standalone GLOMAP-mode model is not currently implemented in the UM."

➢ I highlighted this line because on my initial read through, I was concerned that the assumption of a static cloud pH could impact the results contained within. I saw much later in the discussion that this assumption was addressed- it may be worth mentioning that you address this later in Section XX.

We have endeavoured to address as many of the caveats as possible in the Results section. The following sentence has been added to this paragraph:

"We address the assumption of a fixed pH in the Discussion (Section 4)."

➢ Lines 230-235 and 285-296: I found myself struggling here to translate some of the mathematical statements to readable words. Instead of stating "Ta*TN > K$_p$" or "Ta*TN < K$_p$", I suggest the authors put what this means into physical context using words for the readers. For example: "If available nitrate & ammonium suggest the equilibrium of Reaction 2 is in the forwards direction, thereby promoting the condensation of HNO$_3$ and NH$_3$ to form NH$_4$NO$_3$ (Ta*TN > K$_p$), then the equilibrium concentration of NH$_4$NO$_3$ it is solved using Eq 3. Otherwise…"

We thank the Reviewer for the suggestion. The sentences have been expanded accordingly:

"If the available nitrate and ammonia suggest that the equilibrium of Reaction 2 is in the forwards direction, thereby promoting the condensation of HNO$_3$ and NH$_3$ to form NH$_4$NO$_3$ ($T_A^* T_N > K_p$), then the equilibrium concentration of NH$_4$NO$_3$ is solved using Eq. 3."

➢ Lines 293-295: I was confused about the assumption that ammonium nitrate does not nucleate new particles and thought some elaboration was needed here. There's a recent Nature paper showing that ammonium nitrate out of equilibrium in clouds can be important nucleation/growth events even if such a state is only reached for a few minutes (https://www.nature.com/articles/s41586-020-2270-4). This is likely less important within a global climate model, but it is worth at least a sentence or two of elaboration.

We thank the Reviewer for highlighting this interesting new piece of science. The following elaboration has been added to the paragraph.

"Ammonium nitrate chemistry primarily involves condensation and evaporation (Makar *et al.*, 1998; Benduhn *et al.*, 2016), although Wang *et al.* (2020) have shown that NH$_3$ and HNO$_3$ can condense onto nanoparticles and thus contribute to nucleation events, which may be of importance in urban

settings and at high altitudes. In this model, aerosol number concentrations are not altered explicitly by nitrate chemistry (assuming condensation and evaporation are more important than nucleation)"

The following has been added to the discussion:

Lastly, the assumption that $HNO_3$ and $NH_3$ are only involved in condensation and evaporation and not in nucleation may need to be revisited given developments in the theory of new particle formation (Lee *et al.*, 2019; Wang *et al.*, 2020).

The following references have been added:

Lee, S.-H., Gordon, H., Yu, H. ,Lehtipalo, K., Haley, R., Li, Y., and Zhang, R.: New particle formation in the atmosphere: From molecular clusters to global climate. J. Geophys. Res.-Atmos., 124, 7098–7146. https://doi.org/10.1029/2018JD029356, 2019.

Wang, M., Kong, W., Marten, R. *et al.*: Rapid growth of new atmospheric particles by nitric acid and ammonia condensation. Nature, 581, **184**–189, https://doi.org/10.1038/s41586-020-2270-4, 2020.

> Line 309: Why is alkalinity not titrated for sea-salt? Is it just an assumption that HNO3 is in excess of Na+? I found it odd that this was stated with no reasoning.

This is an assumption inherited from Hauglustaine et al. (2014) based on Fairlie et al (2010), possibly owing to the fact that dust only partially constitutes $Ca^{2+}$, whereas sea-salt is predominantly $Na^+$. References have now been added to the paragraph.

Additionally, dust is assumed to uniformly constitute 5 % $Ca^{2+}$ by mass as in Hauglustaine *et al.* (2014), which differs from the approach in Remy *et al.* (2019) who used a spatially hetereogeneous $Ca^{2+}$ fraction more akin to observations. Dust alkalinity is titrated by uptake of $HNO_3$ until the dust pH is neutralised whereupon $HNO_3$ stops condensing [Fairlie *et al.*, 2010; Hauglustaine *et al.*, 2014], while no such limitation is necessary for sea-salt which generally constitutes a higher fraction of $Na^+$ per mass than dust constitutes $Ca^{2+}$ [e.g. Xiao *et al.*, 2018].

The following reference has been added:

Xiao, H.-W., H.-Y. Xiao, C.-Y. Shen, Z.-Y. Zhang, and A.-M. Long: Chemical Composition and Sources of Marine Aerosol over the Western North Pacific Ocean in Winter, Atmosphere, 9, 298; doi:10.3390/atmos9080298, 2018.

> Line 389: It seems to me that the uncertainty in emissions driving the model are major contributor to the discrepancies between the model and observation comparisons made in the results section – which is why there is time spent making seasonal and spatial comparisons between the model and observation networks. I think it's worth a sentence here preparing the reader for that pointing out the significance of these emissions uncertainties – something along the lines of "Such large discrepancies in NH3 emissions inventories can impact direct model-measurement comparisons which make it important to consider the spatial and temporal trends rather than just the overall magnitudes. For this work exploring the sensitivity of NH4NO3 to thermodynamic equilibrium assumptions, the direct comparison of model performance to observations is done with the goal of understanding the degree to which thermodynamic assumptions may push the model out of realistic behaviour rather than best re-creating the observations".

This is an excellent suggestion by the Reviewer, and we'd like to say thank you for the suggested text which we've adopted in its entirety.

"Such large discrepancies in $NH_3$ emissions inventories can impact direct model-measurement comparisons which make it important to consider the spatial and temporal trends rather than just the overall magnitudes. For this work exploring the sensitivity of $NH_4NO_3$ to thermodynamic equilibrium assumptions, the direct comparison of model performance to observations is done with the goal of understanding the degree to which thermodynamic assumptions may push the model out of realistic behaviour rather than best re-creating the observations"

> ➢ Line 419-420: I wanted to highlight these lines, because I they do a really good job of "summarizing" what the previous paragraph states / contextualizing the results pointed out. It is what I was longing for in other parts of the results section. I suggest the authors look through the other final sentences in the results section and model them after this paragraph.

We have endeavoured to add introductory and/or concluding remarks to all the paragraphs in the Results section, as highlighted in the new manuscript with tracked changes.

> ➢ Lines 421-434: This is one of the results paragraphs that is not well contextualized to the physical chemistry occurring/ the overall point of the results communicated. The authors point out the differences in the fast and slow fine and coarse model NO3 sources and burdens, coarse fractions & how they compare to other models. But they leap to " these results… corroborates the assertion that NH4 and NO3 aerosol production is at the upper end of efficiency & that NH3 is limiting" without clearly explaining why. (e.g. Fine mode production of NO3 is faster in FAST (ok, duh), but coarse mode production of NO3 is faster in SLOW (why physically? Is it that there is more HNO3 in the gas phase because NH4NO3 production is slower, so that more HNO3 is available to condense on the coarse mode particles?) Then contextualize WHY that the % of the NO3 burden in the coarse mode of SLOW is closer in HA14 and FAST is closer to XP12 – what assumptions do those studies make about the HNO3 uptake (instantaneous?). Is the % of NO3 burden in coarse mode comparisons just to say that it's within the range of other models? That's what the final sentence seems to suggest. This just needs to be more clear- there's a sentence missing connecting the stated results to the conclusion made.

The paragraph has been split in 2 and extra detail has been added (indicated by italics below):

"Significant differences between the FAST and SLOW simulations are highlighted by the global $NO_3$ metrics in Table 3. In particular, the fine $NO_3$ source is 6.3 Tg[N] yr$^{-1}$ in FAST but only 2.7 Tg[N] yr$^{-1}$ in SLOW, marking a 57 % decrease. Conversely, SLOW exhibits 5 % more coarse $NO_3$ production than in FAST, *which is likely due to the surplus $HNO_3$ in SLOW owing to the reduced fine $NO_3$ production.* The difference is equally discernible in the burdens, with 47 % of the total $NO_3$ burden as coarse $NO_3$ in FAST compared to 67 % in SLOW. This can be compared to a 72 % coarse fraction in HA14 and 47 % in XP12, *suggesting that the FAST and SLOW coarse fractions are between the instantaneous thermodynamic equilibrium model of HA14 and hybrid dynamical nitrate scheme of XP12. Note though that intuitively the coarse ratio in FAST would be expected to be close to HA14 (given that FAST is indistinguishable from the INSTANT simulation), whereas it is closer to XP12, which is probably due to differences in the precursor gas concentrations between FAST and HA14.*

The total $NO_3$ burdens of 0.2 Tg[N] in FAST and 0.15 Tg[N] in SLOW are commensurate with 0.18 Tg[N] in HA14, 0.17 Tg[N] in XP12, and the AeroCom median of 0.14 Tg[N] in BI17. The $NH_3$ burden in FAST (0.04 Tg[N]) is at the lower end of the AeroCom range in BI17 (0.04 to 0.7 Tg[N]) while the $NH_4$ burden in FAST (0.42 Tg[N]) is at the upper range of BI17 models (0.13 to 0.58 Tg[N]), *suggesting*

*that NH₃ is more rapidly neutralised to aerosol in the UM than in other GCMs. This* corroborates the assertion that NH₄ and NO₃ aerosol production in the UM is at the upper end of efficiency when compared to other existing GCMs and suggests that NH₃ rather than HNO₃ is the limiting factor controlling NH₄.NO₃ production in these simulations*, given that the NH₃ burden in FAST is negligible.* In summary, Table 3 illustrates the close parity with regards global and annual mean metrics between the UM simulations and previous nitrate simulations with various climate models from the literature."

➢ Figure 4: This figure was particularly hard for me to decipher, and I spent far too long trying to understand the message it was communicating. The conclusion in the text is the only reason I was able to follow what this was showing. First, the key has cut off the symbol for "hetNO₃ Fast" so that its symbol looks identical to NO₃ fast- I assume that the solid line is NO₃ fast and the long dashes are hetNO₃, but the key does not distinguish them. I think this plot could be greatly improved if it was more clear which lines we're supposed to be comparing on each figure. Right now, that is done with the different line styles (e.g. compare black solid like to black solid line), but I think the line thickness are different between the fast and slow simulations, which makes identifying which lines are supposed to be comparable difficult (e.g. the dashed black line looks totally different than the color dashed line). Another way to improve this figure would be to improve the key- Put the slow and fast lines that are comparable next to one another (so 2 columns in the key).

We agree with the Reviewer that Fig. 4 is too complex in its current state. Our main aims with Fig. 4 were to highlight the similar seasonal cycles in NO₃ concentrations and NH₃ emissions and to show that the fine NO₃ concentrations in the SLOW simulation are of similar magnitude to the coarse NO₃ concentrations. We have now combined Figs 4 and S7 and removed the map indicating the Giorgi regions, and the Central South America timeseries which had the smallest NO₃ concentration of the chosen regions. The new Figure 4 contains detail that is relevant for the conclusion.

Additionally, we have added Table S6 to the Supplement which gives latitudinal and longitudinal ranges of the Giorgi regions used.

[Figure]

**Figure 4: Regional and monthly-mean NO₃ (solid line) and coarseNO₃ (dashed line) near-surface concentrations and NH₃ emissions (solid black line) time-series for the SLOW (brown) and FAST (green) simulations for 9 'Giorgi regions' [Giorgi, 2006] (land-only) representing high NO₃ concentration areas.**

➢ Line 499-506: Clarity of result could be improved/ needs a better summary sentence.

The following has been added to the end of the paragraph:

"In summary, Fig. 4 shows the tight coupling between regional NH₃ emissions and adjacent NO₃ surface concentrations in many regions and highlights the strong seasonality of NH₄NO₃ in the UM."

➢ Line 518: The authors mention several times that the "model output is linearly interpolated to measurement sites", but it's not totally clear exactly what that means. Is it that value assumed at the specific lat/long of the site is actually interpolated from its distance to other grid boxes & their values?

The interpolation used is actually "nearest neighbour" assuming the Earth is a sphere rather than the "linear" or "bilinear" quoted. The text has been changed in both instances to reflect this.

➢ Line 525-533: This is another section where I was looking for a better summary sentence. I'm not sure if the point of these figures is to show that FAST/SLOW is "better" at simulating observations or that simply changing the assumption about thermodynamic equilibrium has a huge impact on NO₃, NH₄, and HNO₃. Figure 5 and 6 are pushing me to decide which assumption is "better"- but the emissions bias seems to muddle making a direct conclusion. E.g. SLOW looks closer to the 1:1 line, but it's only there because of emissions biases? Does this mean FAST is better? Is that the point of the figure? Perhaps you could add a "theoretical" 1:1 line to show what you'd expect if there weren't emissions biases? The conclusions section describing this result is much clearer (lines 710-715) than the results discussion here.

We agree that the language of these paragraphs and the overall result is too ambiguous. We also think that Fig. 5 is too confusing in its current state. The first change we have made is to combine the East and West measurements in Fig 5 (the separation was not necessary):

[Figure]

Secondly, we have introduced the Figures using a suitable sentence:

"When introducing an aerosol such as $NH_4NO_3$ to a GCM, it is important to validate the model by comparing the simulated concentrations to observations."

Thirdly, we have provided a summary statement for Fig. 5 that explains the difficulty in making a robust conclusion given the error in $HNO_3$.

"Because of the underlying $HNO_3$ bias, it is not possible to declare whether FAST or SLOW is the better model from comparison with the CASTNET observations (Fig. 5). It is only possible to deduce that reducing the $HNO_3$ uptake coefficient ($\gamma$) in SLOW leads to a substantial reduction in total $NO_3$ concentration, as already shown in Figs 1, 2 and 4."

Lastly, we have provided an overall summary statement for Fig 5 (US) and Fig. 6 (Europe):

"In summary, Figs 5 and 6 demonstrate the high skill of the UM nitrate scheme in capturing the magnitude of observed $HNO_3$, $NH_4$ and $NO_3$ concentrations and highlight how the $HNO_3$ uptake coefficient ($\gamma$) could be used to tune $NH_4NO_3$ concentrations in a GCM to observations."

> Line 597-599: I was also wondering about $N_2O_5$ chemistry being promoted in the winter & how that is simulated in the model? Perhaps worth a mention here as a reason for the PM2.5 disagreements. (Maybe see Shah et al., 2018: https://www.pnas.org/content/115/32/8110/tab-article-info).

We thank the Reviewer for highlighting this paper. As mentioned in the discussion, heterogeneous $N_2O_5$ hydrolysis on sulfate surfaces is currently parameterised using a fixed uptake coefficient of 0.1. The relative humidity and liquid water dependent parameterisation used in Shah et al (2018) may well improve the simulation of seasonal nitrate concentrations. We have added the following caveat to the sentence.

"An additional source of uncertainty that may affect the seasonal $NH_4NO_3$ cycles is the dependence of heterogeneous $N_2O_5$ hydrolysis on relative humidity and aerosol liquid water content. In the UM, $N_2O_5$ hydrolysis on sulphate is parameterised using a fixed uptake coefficient of 0.1, whereas Shah *et al.* (2018) have shown that a humidity and acidity dependent uptake coefficient improves $PM_{2.5}$ forecasts in winter over the eastern US."

The following reference has been added to the manuscript:

Shah, V., et al..: Chemical feedbacks weaken the wintertime response of particulate sulfate and nitrate to emissions reductions over the eastern United States, P. Natl. Acad. Sci. USA, 115, 8110, https://doi.org/10.1073/pnas.1803295115, 2018.

> Figure 10 & Section 3.4: I'm not sure that Figure 10 adds anything to the paper that cannot be described in the text. The figure has so many panels that it's very confusing to read and hard to compare the panels needed to get to the in-text conclusions. I suggest the authors consider moving it to the supplement or simplifying the figure.

We have decided to remove Fig. 10, Supplementary Figs S11 and S12, and all of Section 3.4 as they added very little to the central aim of the manuscript which is the sensitivity test on $\gamma$.

> Figure 12: Given the small region where the TOA radiation differences are significant between the two results, I also suggest cutting this figure or moving it to the supplement. The results can be adequately described in the text without a figure.

This figure has been moved to the Supplement (Supplementary Figure S11)

**Reviewer 2**

**General Comments**

> The main overarching issues were pertaining to some detailed explanations in methods and their explanatory text in main manuscript that can either be summarized, tabulated and/or shifted to supplement to allow reader(s) to get at the results and discussions quickly, and not be overwhelmed before they get to that part. The lack of brevity specifically in 'Methods' and to some extent in 'Results' needs to be addressed by the authors before a revised version of manuscript can be accepted finally for publication.

Section S1 (Unified Model configuration) has been added to the Supplement which includes all the details in the Methods that were not directly relevant to the nitrate scheme. The results section has been refined. We have added many introductory remarks and concluding comments to the results section to improve its readability. Additionally, we have simplified 2 figures (Figs 4 and 5) and removed 3 extraneous figures (Figs 9, 10, and 12) from the manuscript. We have also removed Section 3.4 from the manuscript as it contributed little. In our, opinion these refinements have greatly improved the manuscript's readability and scientific reach.

> Also, the distinction between fine vs coarse $NO_3$ in terms of nomenclature say coarse mode nitrate as hetNO3 can be confusing to readers

The abbreviation $hetNO_3$ has been changed to $coarseNO_3$ throughout the manuscript following Reviewer 1's recommendation

**Specific Comments:**

> Lines 40-60: Make sure the paragraphs are Justified instead of left aligned and ensure the same in rest of manuscript consistently.

The paragraphs have been justified

> Lines 122-123: Edit the following as: "The hybrid-dynamical nitrate scheme developed by Benduhn et al. (2016) in the standalone GLOMAP-mode model is currently not implemented in the UM."

The line has been changed as suggested. Thank you for the suggestion.

> Lines 125-126: Edit the following as "…in order address the gaps in modelled NH4 and NO3 with their respective observations."

The line has been changed as suggested. Thank you for the suggestion.

> Section 2.1: Will suggest the authors to consider summarizing the configuration of the UM used to test the new nitrate scheme as described in section 2.1 in tabular form either in main manuscript or even move to supplement. This will help readers to get to the focus of reader(s) quickly to the new nitrate thermodynamic scheme in UM quickly.

Section S1 (Unified Model configuration) has been added to the Supplement which includes all the details in the Methods that were not directly relevant to the nitrate scheme. Associated references have been moved to the Supplement accordingly.

> ➢ Table 1: a) please denote that $\sigma$ is referred to geometric standard deviation in the Table 1 caption.

This has been added to Table 1.

> ➢ Table 1: b) Can the authors enunciate more on the difference between NO3 and hetNO3 in accumulation and coarse mode? (Lines 209-210 [*"NH4 and NO3 mass is emitted into the Aitken and accumulation soluble modes and may be transferred to the coarse soluble mode via aerosol processing, while hetNO3 is limited to the accumulation and coarse soluble modes.*"] seems to explain this? But might help to etch it out more in terms of this question and Table 1).

The following has been added to the discussion on Table 1 at the start of Section 2.2.

"Note that 'NO$_3$' refers solely to NO$_3$ associated with NH$_4$, while 'coarseNO$_3$' refers to NO$_3$ associated with dust and sea-salt."

> ➢ Table 1: c) In Lines 409-410: authors state "'fine NO3' refers to NO3 associated with NH4 while 'coarse NO3' refers to NO3 associated with dust and sea-salt (i.e. NO3 in hetNO3)." Similar clarification to address comment b) early on in the text would be helpful.

Please see the last reply

> ➢ Section 2.2.1: Please ensure chronological ordering of supplemental figures (for instance S2 comes after S3 in text now), re-arrange in the order they are mentioned in text. Make sure Tables, figures are arranged in a chronological order in general throughout the manuscript.

We thank the Reviewer for pointing out this error. We have swapped Figs S2 and S3, and Tables S2, S3 and S4 in the Supplement to ensure they are mentioned in the right order.

> ➢ Lines 279-281: *"However, atmospheric Aitken mode number concentrations generally exceed accumulation mode concentrations, particularly over populous land regions and increasingly with altitude."* Referring to the above statement, did the authors observe any converse trend (i.e. higher accumulation mode concentrations) over say rural or coastal non-urban regions? Discussing the spatial patterns in Fig S4 briefly in 1-2 sentence(s) would be a valuable inference to add from the manuscript's findings.

We have investigated this further and added the following remark to the paragraph:

"Exceptions to this include near the surface over high-latitude maritime regions, Amazonia, and much of Australasia, where Accumulation number concentration exceed Aitken concentrations on an annual-mean basis in the UM."

> ➢ Lines 294-295: Apart from Wang et al., 2020 (Nature) showing NH4NO3 nucleating for new particle formation (NPF), there has been an increasing interest in exploring NPF parametrizations specifically for bridging the model-observation number concentration gaps at high elevations (https://agupubs.onlinelibrary.wiley.com/doi/10.1029/2018JD029356) . It would be good to point this out may be in discussions on future steps (in conclusions), as NPF parametrizations are still an evolving area when it comes to global climate models.

We thank the Reviewer for highlighting these interesting articles. The following elaboration has been added to the paragraph.

"Ammonium nitrate chemistry primarily involves condensation and evaporation (Makar *et al.*, 1998; Benduhn *et al.*, 2016), although Wang *et al.* (2020) have shown that $NH_3$ and $HNO_3$ can condense onto nanoparticles and thus contribute to nucleation events, which may be of importance in urban settings and at high altitudes. In this model, aerosol number concentrations are not altered explicitly by nitrate chemistry (assuming condensation and evaporation are more important than nucleation)"

The following has been added to the discussion:

"Lastly, the assumption that $HNO_3$ and $NH_3$ are only involved in condensation and evaporation and not in nucleation may need to be revisited given developments in the theory of new particle formation (Lee *et al.*, 2019; Wang *et al.*, 2020)."

The following references have been added:

Lee, S.-H., Gordon, H., Yu, H. ,Lehtipalo, K., Haley, R., Li, Y., and Zhang, R.: New particle formation in the atmosphere: From molecular clusters to global climate. J. Geophys. Res.-Atmos., 124, 7098–7146. https://doi.org/10.1029/2018JD029356, 2019.

Wang, M., Kong, W., Marten, R. *et al.*: Rapid growth of new atmospheric particles by nitric acid and ammonia condensation. Nature, 581, 184–189, https://doi.org/10.1038/s41586-020-2270-4, 2020.

> ➤ Lines 307-309 (see next comment as well on Section 4): Further/clearer explanations on the merit of following assumptions would be better: *"Additionally, dust is assumed to uniformly constitute 5 % Ca2+ by mass, which differs from the approach in Remy et al. (2019) who used a spatially hetereogeneous Ca2+ fraction. Dust alkalinity is titrated by uptake of HNO3 until the dust pH is neutralised whereupon HNO3 stops condensing, while no such limitation is applied for sea-salt."*

This is an assumption inherited from Hauglustaine et al. (2014) based on Fairlie et al (2010), possibly owing to the fact that dust only partially constitutes $Ca^{2+}$, whereas sea-salt is mostly $Na^+$. References have now been added to the paragraph.

"Additionally, dust is assumed to uniformly constitute 5 % $Ca^{2+}$ by mass as in Hauglustaine *et al.* (2014), which differs from the approach in Remy *et al.* (2019) who used a spatially hetereogeneous $Ca^{2+}$ fraction. Dust alkalinity is titrated by uptake of $HNO_3$ until the dust pH is neutralised whereupon $HNO_3$ stops condensing [Fairlie *et al.*, 2010; Hauglustaine *et al.*, 2014], while no such limitation is necessary for sea-salt which generally constitutes a higher fraction of $Na^+$ per mass than dust constitutes $Ca^{2+}$ [e.g. Xiao *et al.*, 2018]."

The following reference has been added:

Xiao, H.-W., H.-Y. Xiao, C.-Y. Shen, Z.-Y. Zhang, and A.-M. Long: Chemical Composition and Sources of Marine Aerosol over the Western North Pacific Ocean in Winter, Atmosphere, 9, 298; doi:10.3390/atmos9080298, 2018.

> ➤ Lines 390-400 (and Section 4): Follow up in conclusion on how uncertainty in different emission sources specifically NH3 with overestimated inventory can be a major source of uncertainty. I see points in Section 4 has been made regarding the NH3 and NOx inventory-induced uncertainty and N2O5 chemistry simplification (Lines 735:738: "An accurate NH3 and NOx emissions inventory is vital for a proficient simulation of NH4 and NO3 concentrations. HNO3 concentrations also appear to be overestimated over the western US (Fig. 5) in these simulations, which may emanate from an oversimplification of heterogeneous N2O5 chemistry in UKCA Strattrop1.0, given that the uptake coefficient in

that reaction is uniformly set to 0.1 (Archibald et al., 2020)."]. But similar impacts from say assumptions made in Lines 307-309 (for instance) can be further dissected in conclusions as they are missing.

We thank the Reviewer for the suggestion. The following has been added to the conclusions.

"Other simplifications such as uniformly assuming that mineral dust constitutes 5 % $Ca^{2+}$ per mass and that the alkalinity of sea-salt may be titrated indefinitely may result in errors in coarse-mode $NO_3$ concentrations [Remy *et al.*, 2019]"

> Figure 3: The only fundamental conceptual critique I have on result section is pertaining to Fig. 3: is that the manuscript does not elucidate much on the 'role of Convection' in their version of UM model that can also bring nucleation precursors (NH4, NO3 for instance) from the ground level to the free troposphere? The vertical profile obviously is limited to 0-6 km as NPF or nucleation has not been included (see comment for Lines 294-295). This limitation is essential to be mentioned as stated earlier, albeit as a future step if beyond the scope of current manuscript.

We thank the Reviewer for the suggestion. The following has been added to the conclusions.

"If NPF is found to dominate $NH_4NO_3$ production at higher tropospheric altitudes than condensation-related production, then the dynamics of convective transport of $NH_4NO_3$ precursors will become important."

> Figs 5,6 and 8: The discussions around these figures seem to give similar inferences and can be synthesized together in terms of their summary pointing to similar inferences on FAST introducing positive biases etc. Its understandable than Figs. 5, 6 are annual means and Fig. 8 gives a seasonal variation, but still can be very much summarized together. Or some parts of them can be moved to supplement.

We agree that Figs 5, 6, and 8 show substantially the same result – that FAST exhibits >2x as much NO3 as SLOW. However, they are sufficiently different to all warrant being included. For instance, its important to evaluate a model against observations in different regions (US in Fig. 5 and Europe in Fig. 6). Additionally, its important to look at nitrate in the context of overall aerosol concentrations (PM2.5, Fig. 8).

We've added the following summary line to our analysis of Fig. 6:

"In summary, Figs 5 and 6 demonstrate the high skill of the UM nitrate scheme in capturing the magnitude of observed $HNO_3$, $NH_4$ and $NO_3$ concentrations and highlight how the $HNO_3$ uptake coefficient ($\gamma$) could be used to tune $NH_4NO_3$ concentrations in a GCM to observations."

We've added the following preface to the discussion on Fig. 8:

"As $NH_4NO_3$ is a significant contributor to urban air pollution episodes (Jiminez *et al.*, 2009), it is important to assess the contribution of $NH_4NO_3$ to overall $PM_{2.5}$ surface concentrations using observations for validation."

> Figure 11: Authors can do away with Panels (a),(b), and (e) and their discussions on AOD results for CNTL, FAST and SLOW sensitivity runs, and just show difference maps (c, f). Can move the Panels (a),(b),(d),(e) as separate figure into supplement and summarize the model vs observed AOD in result text, referring to the new supplemental figure. There is a need to

synthesize the result section, in terms of what take-out messages the authors want to stand out to the reader(s).

We have moved Fig. 12 to the Supplement (Fig. S11) as it was not contributing significantly to the paper. Overall, the Results section has been synthesized by the addition of prefaces and concluding remarks for each figure. For instance, for the AOD plot (now Fig. 9), we have added the following introductory remark:

"Atmospheric $NH_4NO_3$ aerosol may have significant radiative implications on a regional basis leading to climate changes (Hauglustaine *et al.*, 2014). It is thus useful to compare the aerosol optical depth and Top-Of-the-Atmosphere (TOA) radiative flux changes in the FAST and SLOW simulations with CNTL to estimate the radiative impact of $NH_4NO_3$."

In terms of panels (a), (b), and (e), we think it is important to validate the baseline simulation using observations before investigating anomalies in experiments (i.e. the nitrate simulations). To explain this, we've added the following sentence to the paragraph:

"The MODIS $AOD_{550}$ is included in Fig. 9 to assess the skill of the CNTL simulation at capturing the observed $AOD_{550}$ distribution."

---

## Author Response (AR2)

**Response to editor for paper: "Exploring the sensitivity of atmospheric nitrate concentrations to nitric acid uptake rate using the Met Office's Unified Model" by A C Jones et al.**

We thank Dr Murphy for the suggestions. The following comments have been addressed:

➢ Lines 43, 599 – Jimenez is misspelled

This has been changed

➢ Lines 58, 61 McDuffie is misspelled

This has been changed

➢ Line 138 – Is it correct to say "…comprises fine NH4 and NO3 aerosol in the Aitken, accumulation and coarse soluble modes…"? Isn't fine aerosol by definition not coarse?

We use 'fine' to describe $NO_3$ associated with $NH_4$ to be consistent with other papers (e.g. Hauglustaine et al., 2014). The actual amount of $NO_3$ and $NH_4$ that ends up in the coarse mode is negligible – it mostly evaporates before growing to this size – so it's appropriate to call this aerosol fine. The following has been added to the text:

"$NH_4.NO_3$ mostly remains in the Aitken and accumulation modes and thus the moniker 'fine' is appropriate."

➢ Line 170 – Do these effective Henry's law constants have units?

The units have been added

➢ Lines 219, 278 – correct 'forwards' to 'forward'

This has been changed

➢ Line 633 – I suggest replacing "imprecisions" with "differences"

This has been changed